# Tracing cancer evolution and heterogeneity using Hi-C

Dan Daniel Erdmann-Pham[1,2,12], Sanjit Singh Batra[3,12], Timothy K. Turkalo[4], James Durbin[5], Marco Blanchette[5], Iwei Yeh [6,7], Hunter Shain[6], Boris C. Bastian [6,7], Yun S. Song [3,8,9] ✉, Daniel S. Rokhsar[4,8,10,11] ✉ & Dirk Hockemeyer [4,8,10] ✉

Chromosomal rearrangements can initiate and drive cancer progression, yet it has been challenging to evaluate their impact, especially in genetically heterogeneous solid cancers. To address this problem we developed HiDENSEC, a new computational framework for analyzing chromatin conformation capture in heterogeneous samples that can infer somatic copy number alterations, characterize large-scale chromosomal rearrangements, and estimate cancer cell fractions. After validating HiDENSEC with in silico and in vitro controls, we used it to characterize chromosome-scale evolution during melanoma progression in formalin-fixed tumor samples from three patients. The resulting comprehensive annotation of the genomic events includes copy number neutral translocations that disrupt tumor suppressor genes such as *NF1*, whole chromosome arm exchanges that result in loss of *CDKN2A*, and whole-arm copy-number neutral loss of homozygosity involving *PTEN*. These findings show that large-scale chromosomal rearrangements occur throughout cancer evolution and that characterizing these events yields insights into drivers of melanoma progression.

Cancer progression is driven by ongoing selection for mutations that endow the evolving cancer cell with a proliferative advantage compared to its direct precursor and the surrounding normal tissue. Genomic studies have significantly increased our understanding of how individual mutations drive cancer progression[1–3]. Somatic copy-number alterations (SCNAs) are also common in cancer, ranging in size from focal alterations (up to several megabases in length) to deletions and duplications that affect entire chromosomes or chromosome arms[4–7]. Such chromosome-arm-scale aneuploidies have been shown to shape tumor evolution and can be correlated with drug responses[8,9].

Although these SCNAs are accompanied by karyotypic changes, chromosomal rearrangements are not directly assessed in the typical copy-number-based screening methods, such as array comparative genome hybridization (CGH), which also cannot detect copy number neutral changes such as inversions and reciprocal translocations[10,11].

The role of chromosomal rearrangements during tumor initiation and early evolution has been particularly difficult to study in solid cancers[12] because at early and premalignant stages of cancer development, the incipient cancer lesions are generally small and intermixed with normal cells from the surrounding tissue. Unlike

[1]Department of Mathematics, University of California, Berkeley, CA 94720, USA. [2]Department of Statistics, Stanford University, Stanford, CA 94305, USA. [3]Computer Science Division, University of California, Berkeley, CA 94720, USA. [4]Department of Molecular and Cell Biology, University of California, Berkeley, CA 94720, USA. [5]Dovetail Genomics, Enterprise Way, Scotts Valley, CA 95066, USA. [6]Department of Dermatology and Helen Diller Family Comprehensive Cancer Center, University of California, San Francisco, San Francisco, CA 94143, USA. [7]Department of Pathology, University of California, San Francisco, CA 94143, USA. [8]Chan Zuckerberg Biohub, San Francisco, CA 94158, USA. [9]Department of Statistics, University of California, Berkeley, CA 94720, USA. [10]Innovative Genomics Institute, University of California, Berkeley, CA 94720, USA. [11]Okinawa Institute for Science and Technology, Tancha, Okinawa, Japan. [12]These authors contributed equally: Dan Daniel Erdmann-Pham, Sanjit Singh Batra. ✉e-mail: yss@berkeley.edu; drokhsar@gmail.com; hockemeyer@berkeley.edu

hematological cancers, methods that rely on metaphase spreads (e.g., the discovery that the "Philadelphia chromosome" is a driver of cancer progression in chronic myelogenous leukemia[13,14]) are not well-suited to solid tumors. Similarly, methods such as spectral karyotyping[15,16] require cell culture and generally can only detect deletions, duplications, and rearrangements larger than 1–10 Mbp. While chromosomal rearrangements can be detected by fluorescence in situ hybridization (FISH), in solid tumors this requires prior knowledge of which rearrangement to look for[17,18]. New whole genome sequencing-based methods can detect and map the breakpoints of chromosomal rearrangements down to a resolution of ~100 bp[19–21] and infer tumor purity, i.e., the percent of cancer cells present in a sample of tumor tissue, by combining detection of copy number alterations with loss of heterozygosity[22–24]. These methods, however, rely on sequencing and mapping mate-pairs that span rearrangement-breakpoints and therefore require relatively high sequencing depth (at least 40x of the cancer genome) for accurate detection[25]. Importantly, this approach generally fails to detect break and fusion points in regions with repetitive DNA sequences[26] such as centromeres or telomeres, which frequently are involved in large-scale chromosomal rearrangement.

High-throughput chromosome conformation capture sequencing, also known as Hi-C[27], provides a new and more sequence-efficient approach to detecting large-scale chromosomal rearrangements. In Hi-C, genomic loci making three-dimensional contact with each other are converted into linked read-pairs by proximity ligation of restriction-digested fixed chromatin[28]. Due to the polymeric nature of chromatin, the majority of contacts occur between loci on the same chromosome, with a signal that decays along the linear sequence. These intra-chromosomal contacts are superimposed on the higher-order three-dimensional folding structure of the genome, which includes features such as alternating open ("A") and closed ("B") chromatin compartments[27]. Large-scale chromosomal rearrangements bring together loci from different chromosomes, or that were far apart on the same chromosome, altering the Hi-C signal in predictable ways. Notably, Hi-C has recently been optimized for formalin-fixed, paraffin-embedded (FFPE) tumor samples as Fix-C[29], allowing these methods to be applied to archived patient samples throughout cancer progression.

Several computational methods have been developed to analyze Hi-C data to infer copy number variation[30–35]. Three prominent methods, HiNT[36], EagleC[37], and hic_breakfinder[38], are widely considered the state-of-the-art for performing this task. However, since the analysis of clinical samples from cancer tissues is complicated by varying degrees of stromal and other normal cell contamination, it is desirable to accurately and robustly detect translocations and infer tumor purity in cancer cells across a wide range of tumor purities. None of these

methods currently infer tumor purity, nor has robustness of inference under low tumor purities been demonstrated.

Here we study genome rearrangements in melanomas, which are well-suited for cancer progression studies since early-stage tumors and their precursor lesions (melanocytic nevi) are routinely excised from the skin of patients and are therefore available for study[39]. These precursors are typically initiated by activating point mutations in the MAP-kinase pathway[40–42]. As the melanoma progresses and invades deeper into the skin, genomic alterations are more often driven by copy number changes and chromosomal rearrangement rather than point mutations from UV exposure[43]. These genomic changes, and subsequent clonal expansions, lead to tumors with multiple distinct cell types. To analyze Hi-C data from tumor samples, we developed HiDENSEC (Hi-C based Direct Estimation of Copy Number and Structural rEarrangements in Cancer cells), a computational framework that allows us to (1) infer the fraction of cancer cells in mixtures of cancer and normal cells (also termed as tumor purity), (2) estimate absolute copy number across the genome in the cancer cells, and (3) detect and ascribe absolute copy number to large-scale structural variants (Fig. 1). A novel feature of our method is correcting the Hi-C signal for covariates including chromatin compartment and GC content, which allows accurate determination of copy number and tumor purity, as well as higher confidence detection of interchromosomal rearrangements[44,45]. We validated HiDENSEC by analyzing samples from in silico and in vitro mixtures of different known genotypes. We then use Hi-C to track the emergence and evolution of structural variants during melanoma progression in three patients, demonstrating the utility of HiDENSEC to identify and characterize genomic rearrangements and copy number changes during melanoma progression. With this approach we identify two alterations disrupting tumor suppressor genes: a novel balanced translocation disrupting *NF1* and whole chromosome arm exchanges resulting in the loss of *CDKN2A* and *PTEN*.

## Results

### Chromatin contacts in cell mixtures are linear superpositions

The application of high-throughput chromatin conformation capture methods to FFPE samples (Fix-C) opens new possibilities for studying chromosome rearrangements in solid cancers[29]. In Fix-C or Hi-C (we use the terms interchangeably), three-dimensional contacts are readily displayed as a "chromatin contact map" (or matrix), in which the intensity at a point (*x*, *y*) is proportional to the number of read-pairs linking two positions *x* and *y* in the genome. Intra-chromosomal contacts appear as a diagonal band in the contact map of a chromosome with itself, with a characteristic pattern that decays on a megabase scale with increasing distance between genomic loci. Inter-

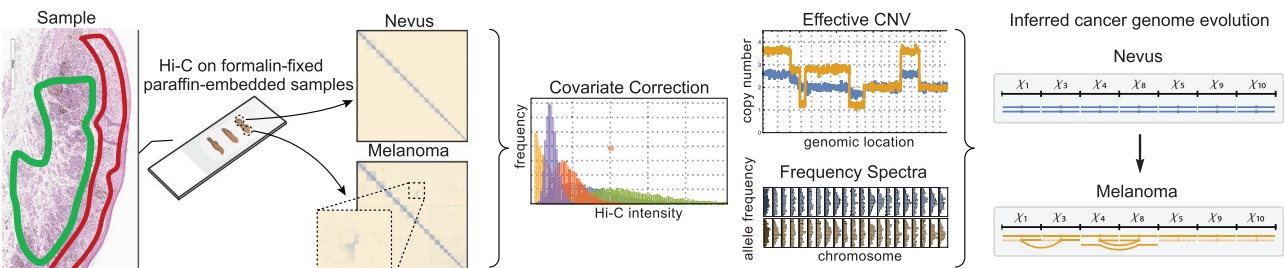

**Fig. 1 | Schematic of HiDENSEC pipeline.** Pipeline (left to right) begins from formalin-fixed paraffin-embedded tumor samples subjected to Hi-C. FFPE samples may be microdissected; the green outline in the example outlines the nevus and the red curve the melanoma area. When aligned to the human reference genome Hi-C reveals large-scale structural variants as off-diagonal enrichments in contact maps. HiDENSEC first corrects the on-diagonal intensities of contact maps for covariates such as chromatin compartments, mappability, GC content, and restriction site

density. These corrected intensities provide absolute copy numbers for every genomic region. Copy numbers for large-scale structural variants inferred from the Hi-C contact maps can also be assigned. Hi-C reads can also be used to compute the regional allele frequency spectrum of germline mutations, which aid in providing higher resolution inference. Combining the output of HiDENSEC on multiple samples from a single patient allows inferences on the temporal order in which structural and copy number alterations during tumor evolution.

chromosomal contacts are generally weak, and show a characteristic open/closed (A/B) pattern[27]. Rearrangements that bring together chromosomal segments that are distant on the reference genome, however, appear as additional "off-diagonal" (i.e., between-chromosome) signals in the Hi-C contact map relating juxtaposed segments. In HiDENSEC we exploit these signals to infer somatic copy number alterations, large-scale chromosome rearrangements, and tumor purity, i.e., the fraction of cells in the tumor sample with each cancer genome type.

To analyze samples that are mixtures of genetically distinct subpopulations of cells, we assume that the chromatin contact map derived from such a heterogeneous sample constitutes the weighted superposition of the contact map of the genomes from individual cells of each subpopulation, with minimal contribution of spurious signals caused by ligation of DNA between neighboring cells (Online Methods). We tested this superposition hypothesis in a synthetic sample generated from a 1:1 mixture of human and mouse cells, which was subsequently fixed with formalin and embedded in paraffin before processing with the Fix-C protocol (Online Methods). We found that the number of Fix-C read pairs connecting human and mouse was less than 0.3% (Supplementary Data 1), which confirms that intercellular proximity ligation is negligible. We can therefore interpret Fix-C contact maps as superpositions of the contact maps of the cell populations within the sample.

## Estimating absolute copy number and tumor purity

We estimate the relative copy number along the genome from Hi-C data, specifically, from the on-diagonal intensities of the chromatin contact matrix computed in 50 kb windows. Each on-diagonal entry measures the total contact frequency of a genomic window with itself, which is nominally expected to be proportional to the copy number of that genomic window. Raw on-diagonal intensities of Hi-C data derived from cancer cell lines, however, empirically show broad distributions that are not easily translated to absolute integer-valued copy numbers along the genome. Since we are specifically interested in describing samples comprising subpopulations with distinct genomic alterations, it is important to obtain quantitative measures of copy numbers.

We reasoned that, in addition to copy number variation, the on-diagonal intensities are likely also influenced by factors such as the density of restriction enzyme cut sites, short-read mappability, sequencing bias due to GC content, and possible effects of variable chromatin compaction along the genome, such as A/B (open/closed) chromatin compartments[27,36,46]. We assessed the impact of these factors on on-diagonal intensities using the GM12878 cell line (which has no copy number variation at this scale), and found that they indeed contributed significantly to overall variation in raw on-diagonal intensity (Supplementary Figs. 1 and 2). Covariate corrections for (1) chromatin compartments, (2) restriction enzyme site density, (3) short-read mappability and (4) GC content explained 80-90% of the variation in signal intensity along the diagonal. We applied this covariate correction to improve copy number inference from Hi-C data (Supplementary Note 1).

Estimating the tumor fraction of each genotype depends critically on accurate inference of absolute copy number profiles. Since (1) the contact map of a cell mixture is the weighted superposition of the contact maps of each subpopulation of cells (as shown above), and (2) the copy number of each region of the genome of an individual cell must be an integer, we jointly infer the cancer cell fraction $f$ and cancer genome copy number profile from mixed samples. The remainder of the sample is generally assumed to be normal cells with frequency $1-f$. In general, however, there may be multiple cancer cell populations with tumor cell fractions $f_1, f_2$, etc. To infer absolute copy numbers and tumor purities from covariate-corrected relative copy number profiles, some prior knowledge about absolute copy numbers is necessary (Supplementary Note 1). By default, HiDENSEC assumes knowledge of the predominant copy number profile in a given sample, which is typically the normal diploid profile; alternative specifications can be incorporated as optional inputs to HiDENSEC.

We validated our absolute copy number and tumor purity estimation method using mixtures of karyotypically normal GM12878 cells and HCC1187 cancer cells in two ways: (1) in silico mixtures of Hi-C data from the two cell lines, and (2) Fix-C data derived in vitro from pelleted, FFPE-fixed mixtures of cells with known karyotypes[47]. Analysis of Hi-C data from these in silico and in vitro mixtures, and its comparison with copy number characterizations obtained from methods based on whole genome sequencing, confirmed the accuracy of the absolute copy number estimates of the cancer cell subpopulation, and the accuracy of tumor purity estimates (Fig. 2a–d and Supplementary Fig. 3a). Additional evidence for HiDENSEC's external consistency is obtained from comparing Hi-C inferences on patient data (see following section) with their associated UCSF500 gene panel data (Supplementary Fig. 5). Although these experiments involve at most two subpopulations within a given sample, resolving more heterogeneous mixtures is governed by the same parameters (effective copy number changes relative to noise). Therefore, the resolution thresholds identified in these two-population situations are expected to equally apply to multi-populations contexts.

We note that in the most general setting the problem of estimating tumor purity using bulk data from samples that are complex mixtures of normal and multiple types of cancer cells is underdetermined. For example, the Hi-C map of a mixture of normal cells and a single subpopulation with many translocations cannot be distinguished from a map that arises from the superposition of many subpopulations (one for each translocation), as long as each subpopulation's mixture proportions are similar. Any method attempting to call subpopulations must resolve this ambiguity in some manner. HiDENSEC appeals to parsimony, and seeks to characterize the smallest number of subpopulations that are statistically compatible with observed copy number profile (Supplementary Note 1). This in turn depends on the signal-to-noise at the given sequencing depth; higher depth may allow two cell populations with similar cell fractions to be distinguished. For the samples considered here, we find that one or two cancer genotypes (along with the diploid reference) are sufficient to account for our data, although we cannot rule out additional low-frequency clonal sub-populations, as observed in other contexts[48–50].

## Detecting reciprocal and copy-number-altering translocations

"Off-diagonal" signals of the Hi-C matrix represent read-pairs arising from contacts between distant regions of the genome, either far apart on a single chromosome or between different chromosomes. As part of HiDENSEC we developed automated methods for detecting such rearrangements, including (1) rearrangements whose breakpoints coincide with copy number changes (type-1) and (2) reciprocal translocations, which are copy-number neutral but show a characteristic "bow-tie" pattern in the Hi-C contact map (type-2). In both scenarios, each candidate rearrangement is associated with a p-value, allowing assessment of significance through standard multiple testing procedures (Online Methods). We also found a small number of alterations that were neither type-1 nor type-2 (Supplementary Data 3). In particular, the more intensive chromatin contacts in smaller chromosomes make some rearrangements difficult to detect in an automated fashion, and these were identified by manual curation of off-diagonal Hi-C signals.

The absolute read counts arising from inter-chromosomal contacts from different cells satisfy the superposition principle (Fig. 2e, f) but vary substantially in magnitude between individual translocations (Supplementary Fig. 4). We therefore only used the presence/absence of these off-diagonal signals to detect large structural variants, and inferred absolute copy numbers and tumor purity based on the on-diagonal intensity analysis as described above (Supplementary Note 1).

 

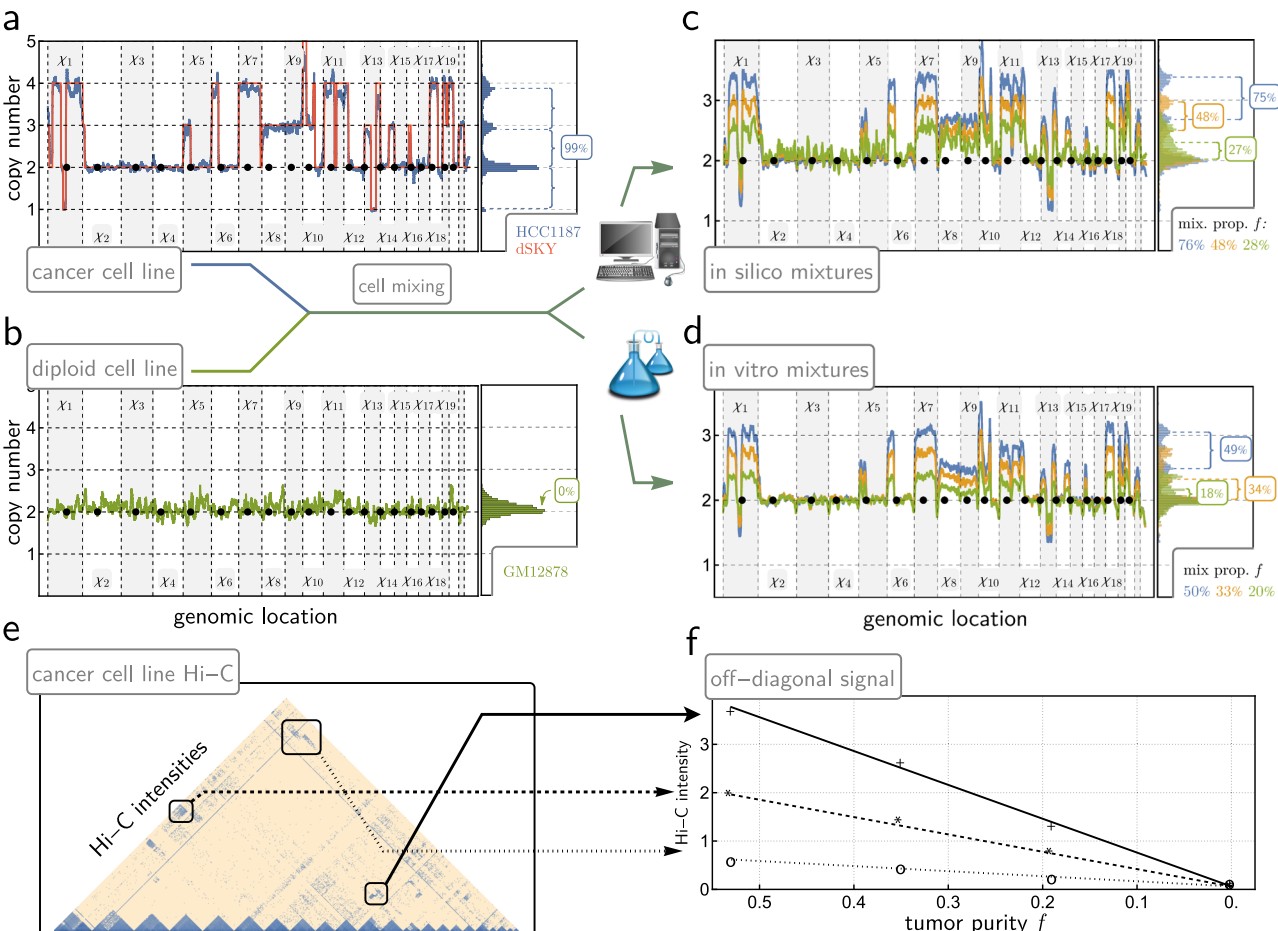

**Fig. 2 | Validation of HiDENSEC using mixtures of samples. a** The HiDENSEC absolute copy number inferences for the cancer cell line HCC1187 are overlaid with copy numbers inferred using dSKY[47]. The horizontal axis represents genomic position, with the alternating gray and white bands representing odd and even chromosomes, respectively. The i-th chromosome is denoted by $\chi_i$. A histogram of absolute copy numbers is aligned on the right, illustrating the resulting spacings between inferred copy number levels and their relation to the fraction of cancer cells within a given sample. **b** The HiDENSEC absolute copy number inferences for the purely diploid GM12878 cell line. The histogram of the absolute copy number values on the right indicates that this control sample contains no detectable sub-population of cells with DNA copy number changes. **c** Using in silico mixtures of Hi-C data from the HCC1187 cancer cell line, and the purely diploid GM12878 lymphoblastoid cell line, HiDENSEC simultaneously accurately infers tumor purity and genome-wide absolute copy number (ploidy). The blue, orange, and green lines

correspond to mixtures of 76%, 48%, and 28% reads coming from HCC1187. The resulting inferred tumor purities of 75%, 48% and 27% within 2% of their true fraction in the mixture. **d** Using in vitro Fix-C samples from mixtures of the HCC1187 cancer and GM12878 wild type cell lines, HiDENSEC successfully infers tumor purity and genome-wide absolute copy number. The blue, orange, and green lines correspond to 50%, 33%, and 20% HCC1187 cells. The resulting HiDENSEC tumor purities are 49%, 34%, and 18%, respectively, which are all again within 2% of the ground truth tumor purities. Supplementary Fig. 3a depicts the 95% confidence intervals associated with these HiDENSEC tumor purity inferences. **e**, **f** After covariate correction and rescaling (Supplementary Note 1), Hi-C intensities are proportional to tumor purity for three different large structural variants in in vitro mixtures of HCC1187 cancer cells and karyotypically diploid GM12878 cells at varying proportions, confirming that HiC data provides a reliable signal for inferring tumor purity.

We did not explore the magnitude of the off-diagonal signals, which are likely due to changes in chromatin packing after chromosomal rearrangement (Supplementary Note 1).

We validated HiDENSEC's performance on the synthetic in vitro mixtures of cells with known karyotypes as described above (Fig. 3a, b), as well as a melanoma whose Fix-C contact map was annotated manually (Fig. 3c and Supplementary Data 3). For each sample, HiDENSEC performance was assessed relative to HiNT, EagleC and hic_breakfinder by recording top-$k$ recall curves for $k$ up to 60 (i.e., how many of the known rearrangements are recovered among $k$ highest scoring features for each method). We measured recall relative to (1) all genomic rearrangements present, and (2) those belonging to type-1 and type-2 as described above. As indicated in Fig. 3a, for all samples, HiDENSEC achieves higher recall while returning fewer total off-diagonal calls (i.e., fewer false positives) than HiNT. Moreover, its comparison with EagleC and hic_breakfinder reveals three directions in which

HiDENSEC provides meaningful contribution alongside and beyond these two methods:

1. While EagleC and hic_breakfinder's precision in calling rearrangement events rivals or exceeds that of HiDENSEC for cases where the tumor populations is present at relatively large mixture proportions (50% and higher), this trend reverses for smaller sub-populations (30% and lower), where HiDENSEC matches or outperforms the two methods.

2. This emphasis on precision comes at a trade-off in recall: both EagleC and hic_breakfinder call events highly conservatively, resulting in 30% (for large mixture proportions) to 80% (for small mixture proportions) of true events being missed by these methods under default significance thresholds. Relaxing significance thresholds for these methods relative to what their authors suggest resulted in little change to the number of called events. HiDENSEC, on the other hand, achieves consistently high recall in all situations, at the cost of calling two to three times as many

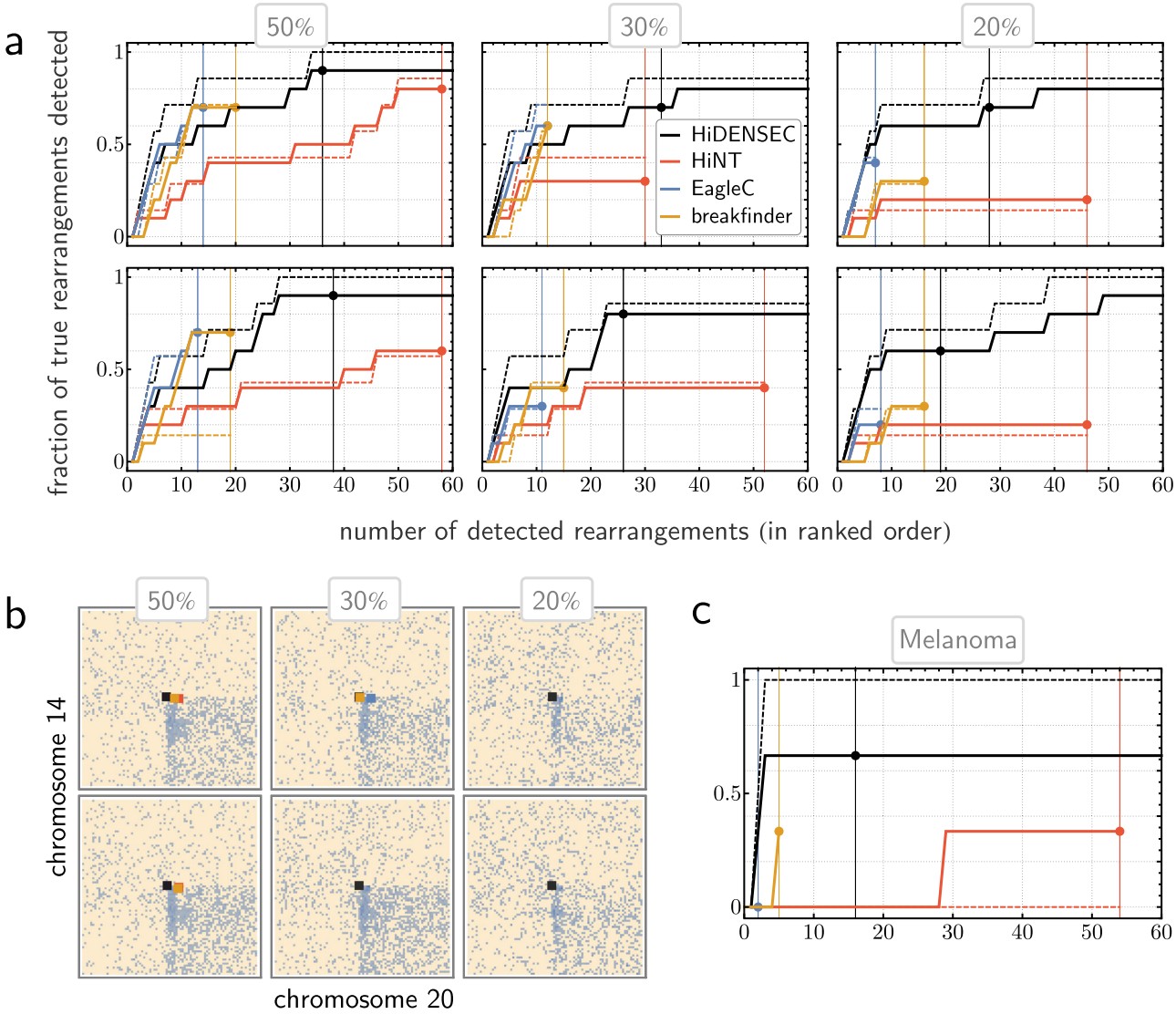

**Fig. 3 | Benchmarking HiDENSEC's identification of genome rearrangements.**
**a** Comparison of HiDENSEC's performance relative to HiNT-TL, EagleC and hic_- breakfinder on the same in vitro mixtures as in Fig. 2 (the second row comprises technical replicates of the first row). Each graph measures top-*k* recall; that is, for each value of *k* (horizontal axis), it indicates the proportion of true large-scale genome rearrangements (as assessed by manual annotation of the Hi-C map or reported in the literature[66]) contained within the *k* most significant calls returned by the respective algorithm (vertical axis). This visualization differs from typical ROC plots, and allows one to read off both recall (vertical axis) and precision (as the fraction of step-increases up to a fixed number of calls). Black and red points on the graphs and their corresponding vertical lines characterize algorithm-specific

significance thresholds, while solid and dashed lines distinguish performance relative to the full set of rearrangements (solid) and relative to the set of only those events classified as either type-1 or type-2 (see main text for definitions).
**b** Illustration of detection thresholds and localization accuracy on a specific type-1 rearrangement (as described in the main text) involving a fusion of a region of chromosome 14q with a region of 20p. Hi-C sub-matrices corresponding to the region of interest (in row-wise arrangement mirroring part (**a**)) are annotated by HiDENSEC's, HiNT's, EagleC's and hic_breakfinder's relevant calls (coloring as in (**a**)). Absence of certain colored squares indicates cases where the associated method does not localize any breakpoints within 2.5 Mb of the true fusion event.
**c** Same comparison as in (**a**) performed on a sample from Patient 4.

events. Either behavior might be preferable depending on the application at hand.

3. The type of events HiDENSEC is designed to detect (large-scale genomic rearrangements associated with copy number variation) appear to be less reliably identified by EagleC and hic_breakfinder, suggesting that one of those methods in tandem with HiDENSEC may deliver complementary performance.

In this analysis we consider an off-diagonal call to be a true positive if the corresponding chromosome pair is joined in the manually curated catalog. To illustrate HiDENSEC's performance in localizing such fusions, Fig. 3b shows an example event that is detected by HiNT at mixture proportions 50% but not below, and by EagleC and hic_- breakfinder for one sample at mixture proportion 30%, but not for its

replicate nor for smaller mixture proportions. This benchmarking analysis focused solely on HiDENSEC's capability to detect transloca- tions rather than its inference of mixture proportions (which we vali- dated separately above), as neither HiNT, EagleC nor hic_breakfinder implement functionality for identifying such proportions.

Though benchmarking based on manually curated translocation events may appear subjective, we found such an approach to be more accurate than comparisons using existing ground truth labels. More specifically, though existing ground truth labels appear to be of high precision, it is not clear that they are exhaustive enumerations of all large-scale rearrangements in the genome in question. For example, K562 (one of the most comprehensively and recently characterized cell lines for which data is available) exhibits at least two intensity patterns

in its Hi-C matrix (Supplementary Fig. 19) that are called by at least one method (HiDENSEC, EagleC, hic_breakfinder or HiNT), and which are strongly indicative of true translocations (indeed, omission of these events leads to inconsistencies of the copy number profile; in particular, they are not attributable to indirect joins involving three or more chromosomal segments which could lead to spurious interaction patterns). Yet neither of these two translocations are included in the external K562 ground truth[51]. The probability of such intensity patterns occurring by chance is negligible; coupled with the ability of Hi-C to directly probe physical proximity of two genomic locations this casts doubt on how useful existing ground truth labels are for high-resolution method benchmarking. We therefore chose manual curation (which produced a set of labels that is a superset of externally provided ones) to obtain benchmarks that we believe are more representative of performance, instead of comparing to available benchmarks of uncertain completeness. Our benchmark set may be useful for other studies, and is provided in Supplementary Data 3.

An alternative means to obtain Hi-C maps arising from known genomic rearrangements would be to use simulation pipelines like FreeHi-C[52] and Sim3C[53]. However, matrices produced by Sim3C appear to generally lack complexities that are typical of real data, including intensity decays and both on- and off-diagonal confounding (Supplementary Fig. 20). The suitability of these simulated maps for benchmarking is thus unclear. FreeHi-C attempts to correct for these simplifications, but does not yet model structural variants of a given genome.

## Using HIDENSEC to reveal the evolution of chromosomal aberrations during melanoma progression

We used HiDENSEC to characterize genome evolution during cancer progression in three patients with melanoma. In all, we generated Fix-C data from eight FFPE samples, representing different stages of melanoma progression (Supplementary Data 2):

(1) Patient 1. A primary cutaneous melanoma (Sample 1 - II) along with an adjacent precursor nevus (Sample 1 - I) in the same tissue sample.

(2) Patient 2. A primary cutaneous melanoma (Sample 2 - I) along with a metastasis that arose later during progression (sample 2-II).

(3) Patient 3. Two histologically distinct regions (1 and 2, Samples 3 - I and 3 - II) from the same primary tumor of an acral melanoma along with a metastasis that arose later (Sample 3 - III).

For each patient, we sequenced and analyzed Fix-C libraries prepared from FFPE sections that were consecutive to sections previously used for targeted short-read sequencing of either a panel of cancer-associated genes (UCSF500[54]) or the entire exome. These data allowed absolute copy number profiles inferred by HiDENSEC to be compared to phylogenetic relationships between progression stages derived from somatic mutations to develop a comprehensive view of the cancer genome.

### Patient 1

HiDENSEC analysis of Fix-C data from the nevus (Sample 1 - I) and adjacent melanoma (Sample 1 - II) from patient 1 revealed balanced translocations between chromosome 4 and 8 and chromosome 1q and 3q, which were only present in the melanoma (Fig. 4a, b). Chromosomal breakpoints for these translocations did not overlap genes known to be involved in melanoma progression (Supplementary Data 3). Chromosome arms 1p and 3p showed reduced copy number with discrete changes in copy number at the translocation breakpoints, indicating that the reciprocal derivative chromosome was lost in the melanoma. In addition, there were copy number losses of chromosomes 5, 9, and 10 (Fig. 4c) estimated by HiDENSEC to represent monosomies, with a cancer cell fraction $f = 57\%$ (Supplementary Data 4). This estimate is consistent with the allele frequencies

determined from the UCSF500 cancer gene panel (Supplementary Data 4 and Supplementary Fig. 5). Somatic variant calling using exome sequencing of the nevus (Sample 1 - I) and the adjacent melanoma (Sample 1 - II) along with a matched normal sample, identified the *BRAF* V600E mutation as a driver mutation present in both the nevus and melanoma (Fig. 4d and Supplementary Fig. 6a). Together, these combined analyses show that our method can detect chromosomal rearrangements and copy number changes in tumor samples with a considerable presence of normal cells (Fig. 4e).

### Patient 2

For patient 2 we compared the primary melanoma (Sample 2 - I) to its subsequent metastasis (Sample 2 - II). While some translocations and copy number changes were shared by both samples, others were unique to the metastasis or primary melanoma (Fig. 5a, b, e). The existence of shared structural variants between the two samples implies that the metastasis arose from a common ancestor with the primary melanoma (Fig. 5e). Our HiDENSEC analyses are consistent with a model in which each sample has a single dominant (but karyotypically distinct) cancer cell population mixed with normal cells. This dominant population comprised $f = 71\%$ of the cells in the primary melanoma and 59% in the metastasis (Fig. 5c and Supplementary Data 4). The cancer cell fraction estimated for the primary melanoma sample is consistent with fraction estimated using the mutant allele frequency for *BRAF* V600E, the presumed initiating oncogene (Fig. 5d) (Supplementary Data 4). Copy number profiles estimated using HiDENSEC are highly concordant (Supplementary Fig. 5) with profiles derived from the UCSF500 capture panel from prior sections of the same tissue area (Supplementary Data 4). As with Patient 1, HiDENSEC analysis applied to Fix-C data detected and characterized translocations that would likely not be detected by array CGH or standard sequencing methods (Supplementary Data 3). For example, both the primary melanoma and its metastasis carry a complex translocation event involving chromosome 2, 5, and 10 (Supplementary Fig. 7a), which is concurrent with loss of 5q, a part of 2p and a part of 10p suggesting that the underlying structural rearrangement occurred early in the primary melanoma, since this is a chromosomal rearrangement shared by both the primary melanoma as well as the metastasis. Moreover, HiDENSEC detects a metastasis-specific translocation between chromosome 11 and 17 that provides a mechanistic context for the copy number loss of 11q and a gain in copy number of 17q in the metastasis. Similarly, we detected a chromosome translocation between chromosome 1 and 15 that is present in the primary melanoma and continued to evolve by fusion with chromosome 13. This analysis highlights HIDENSEC's capacity to deconvolve chromosome scale events during cancer evolution (Fig. 5e).

### Patient 3

The acral melanoma of Patient 3 has been previously characterized by exome and RNA sequencing[55]. Acral melanomas are known to be enriched for structural rearrangements[56]. We analyzed two histopathologically distinct subregions of the primary melanoma and one from the metastasis, which arose years later (Fig. 6). HiDENSEC analysis revealed genetic heterogeneity within the primary tumor with some of the chromosomal alterations passed to the metastasis. Since these Fix-C samples were sequenced more deeply we were also able to characterize allele frequencies of inherited variants and trace haplotype copy number (Online Methods, Fig. 6e). The allele frequency spectra provided independent corroboration of copy number estimates and allowed inference of lost and/or duplicated haplotypes (Fig. 7a). In conjunction with copy number estimates, analysis of the allele frequency of these germline variants allowed us to characterize mechanisms of copy number neutral loss of heterozygosity (Fig. 7a).

The two regions of the primary melanoma shared copy number changes and structural rearrangements (Sample 3 - I and Sample 3 - II),

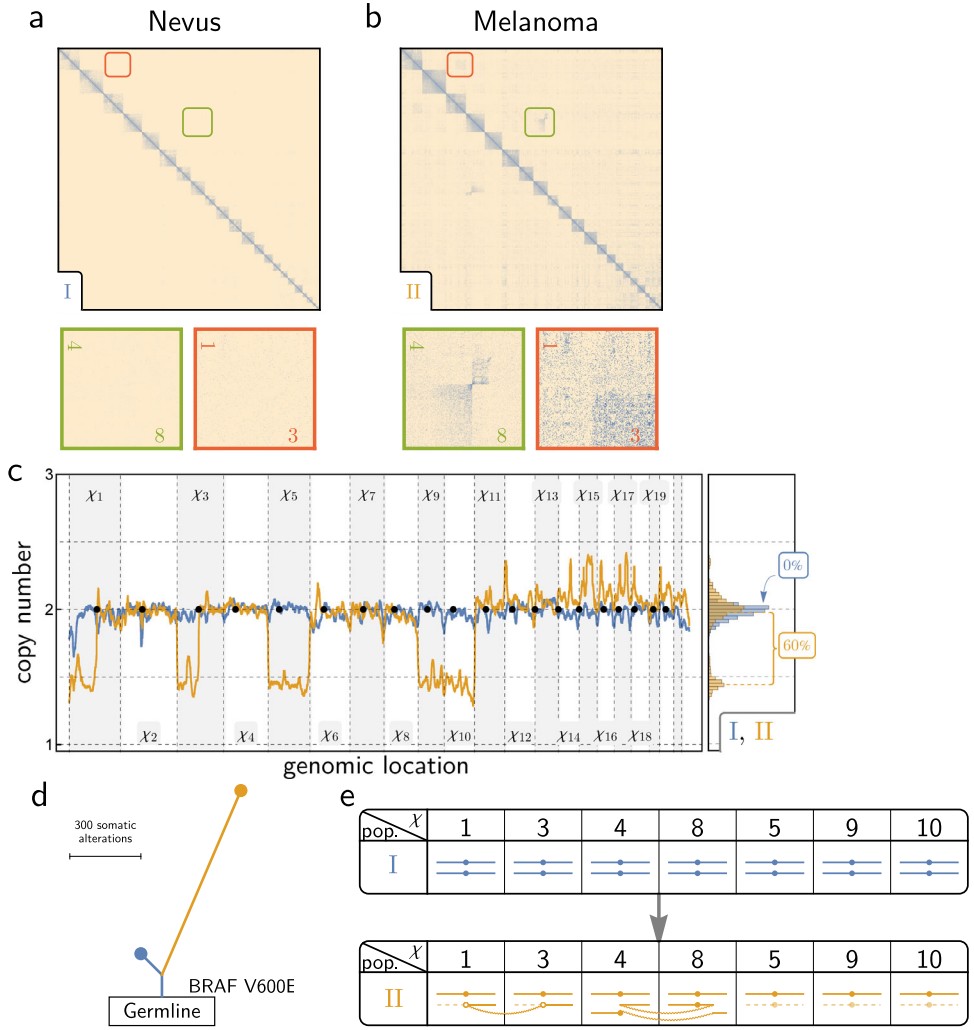

**Fig. 4 | HiDENSEC analysis of Patient 1. a, b** Hi-C maps from the nevus area (Sample 1 - I) and the adjacent melanoma area (Sample 1 - II) are shown along with insets zooming into two different structural variants exclusive to the melanoma area. **c** HiDENSEC absolute copy numbers inferred for both samples (Sample 1 - I in blue and Sample 1 - II in orange) are shown in the same format as Fig. 2a–d. **d** Somatic mutation analysis from exome sequencing yields a phylogenetic tree with *BRAF* V600E as the driver mutation found in both the nevus and the melanoma, but not in the normal control tissue. The length of the branches and the trunk in this phylogenetic tree are scaled based on the number of somatic variants, as described in Supplementary Fig. 6a. **e** Schematics of the inferred karyotypes of the nevus (Sample 1 - I in blue) and melanoma (Sample 1 - II in orange) from Patient 1. Each column represents a chromosome, with the column header denoting the chromosome number. The dashed lines indicate lost fragments, while curved lines connect parts involved in rearrangements. White dots indicate uncertainty about the centromere involved in a rearrangement.

indicating that they are clonally related (Supplementary Data 5). The quantitative increases in copy number of chromosome arms 1q and 6p, and the decreases in copy number of chromosomes/arms 6q, 9, 10, 11p, and 21, in Sample 3 - I and Sample 3 - II, however, requires the presence of more than one cancer cell population (Fig. 6d and Supplementary Note 1). Allowing two cell types A and B, we find that Sample 3 - II comprises a simple mixture of 44% normal cells and $f_A = 56\%$ cancer cells with genome A (Supplementary Data 5). Knowledge of cancer genome A then allows us to interpret Sample 3 - I as a mixture of normal and cancer genome A cells with a second cancer cell population with genome B. We inferred that Sample 3 - I comprises $f_A = 60\%$ cancer cells with genotype A, $f_B = 12\%$ cancer cells with B genotype, and $1 - (f_A + f_B) \approx 28\%$ normal cells (Supplementary Data 5). Finally, the metastatic sample (3 - III) can be described as a mixture of normal cells and a third cancer cell population with genotype C with a cancer cell fraction $f_C = 63\%$. As discussed below, genome C is closely related to genome A of the melanoma (Fig. 7).

The karyotypes of cancer cell genotypes A, B, and C as inferred by HiDENSEC are shown schematically in Fig. 7a, b. These genomes exhibit shared and unique copy number gains/losses and large-scale rearrangements to varying degrees reflecting the phylogenetic relationships among the three cancer genomes. Based on a parsimony analysis of multiple shared chromosome-scale features of cancer cells with genotypes A and C, we infer that they share a more recent common ancestor, with melanoma genome B diverging earlier, and parsimonious reconstructions of the AC and ABC ancestors are also shown in (Fig. 7b and Supplementary Fig. 8). This phylogeny of the three cancer cell genomes provides a framework for understanding changes in karyotype through cancer progression.

Allele frequency spectra derived from the tumor Fix-C data (Online Methods) are consistent with copy number changes inferred by HiDENSEC and provide additional information about genomic changes during cancer progression (Fig. 6e). While most diploid chromosomes have the expected 1:1 ratio of reference:alternate alleles,

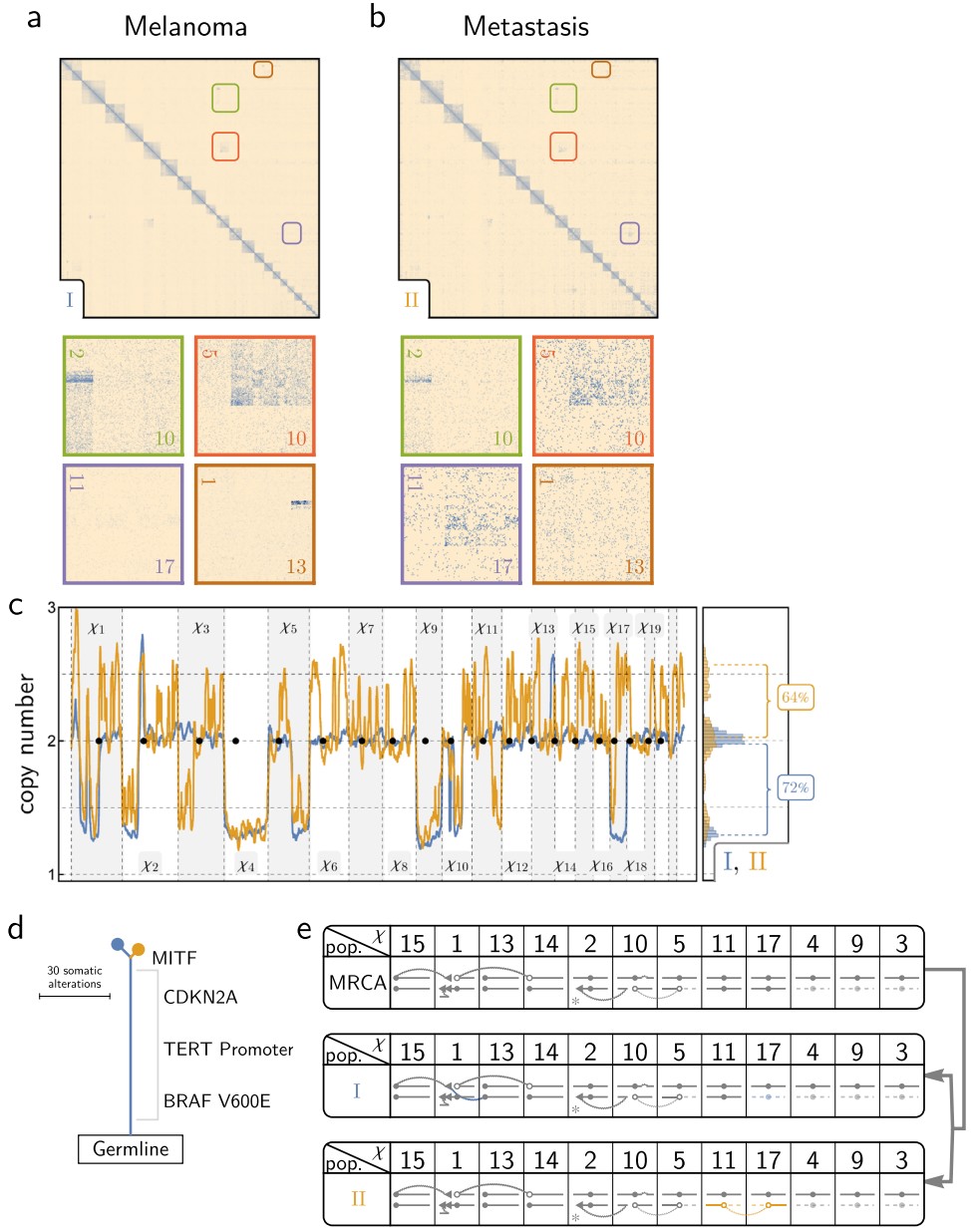

**Fig. 5 | HiDENSEC analysis of Patient 2. a, b** show the Hi-C maps from the primary melanoma (Sample 2 - I) and the corresponding metastasis (Sample 2 - II), respectively. The insets zoom into large-scale structural variants that are observed in the two samples. **c** HiDENSEC inferred absolute copy number for the two tumors along with the inferred tumor purities. **d** Somatic mutations from UCSF500 cancer gene panel sequencing yield a phylogenetic tree with *BRAF* V600E, a *TERT* promoter mutation and a *CDKN2A* mutation as some of the driver mutations. The length of the branches and the trunk in this phylogenetic tree are inferred using the somatic allele frequencies of all somatic variants, as described in Supplementary

Fig. 6b. There is a metastasis-specific somatic mutation in *MITF* which is known to be associated with melanoma progression, and the loss of the p-arm of chromosome 3 and the wild-type allele in the metastasis. **e** Schematic of the various structural variants observed in the two samples and the inferred genome of their most recent common ancestor (MRCA), with a phylogeny shown to the right. Notation mirrors that of Fig. 4e, with additional triangles indicating inversion events. *The translocations between chromosomes 2, 5, and 10 are elaborated upon in Supplementary Fig. 7a.

in some cancer cells we found copy number neutral loss of heterozygosity, indicated by the presence of two copies of the same haplotype (depicted as chromosomes with the same color in Fig. 7a). Chromosomes or chromosome arms predicted to be mono- or trisomic by HiDENSEC show corresponding deviations from 1:1 allele ratios.

For example, trisomic chromosomes such as chromosome 7 (Fig. 6e) show allele frequencies consistent with a 1:2 allelic ratio for genotypes A and C (with observed signals diluted by the fraction of

wild type cells in the sample; chromosome 7 is diploid in genotype B). The chromosome 7 haplotype that duplicated in the AC ancestor carries an oncogenic *BRAF* N581I mutation (Fig. 7a, c). This mutation must have arisen early in cancer progression, because it is found in all three cancer cell genotypes, consistent with its appearance in targeted sequencing[55]. Our HiDENSEC analysis shows that the copy number increase in this mutation in the AC progenitor was associated with duplication of the entire chromosome carrying the mutation (Fig. 7a). Our analysis also establishes the haplotype on which the mutation

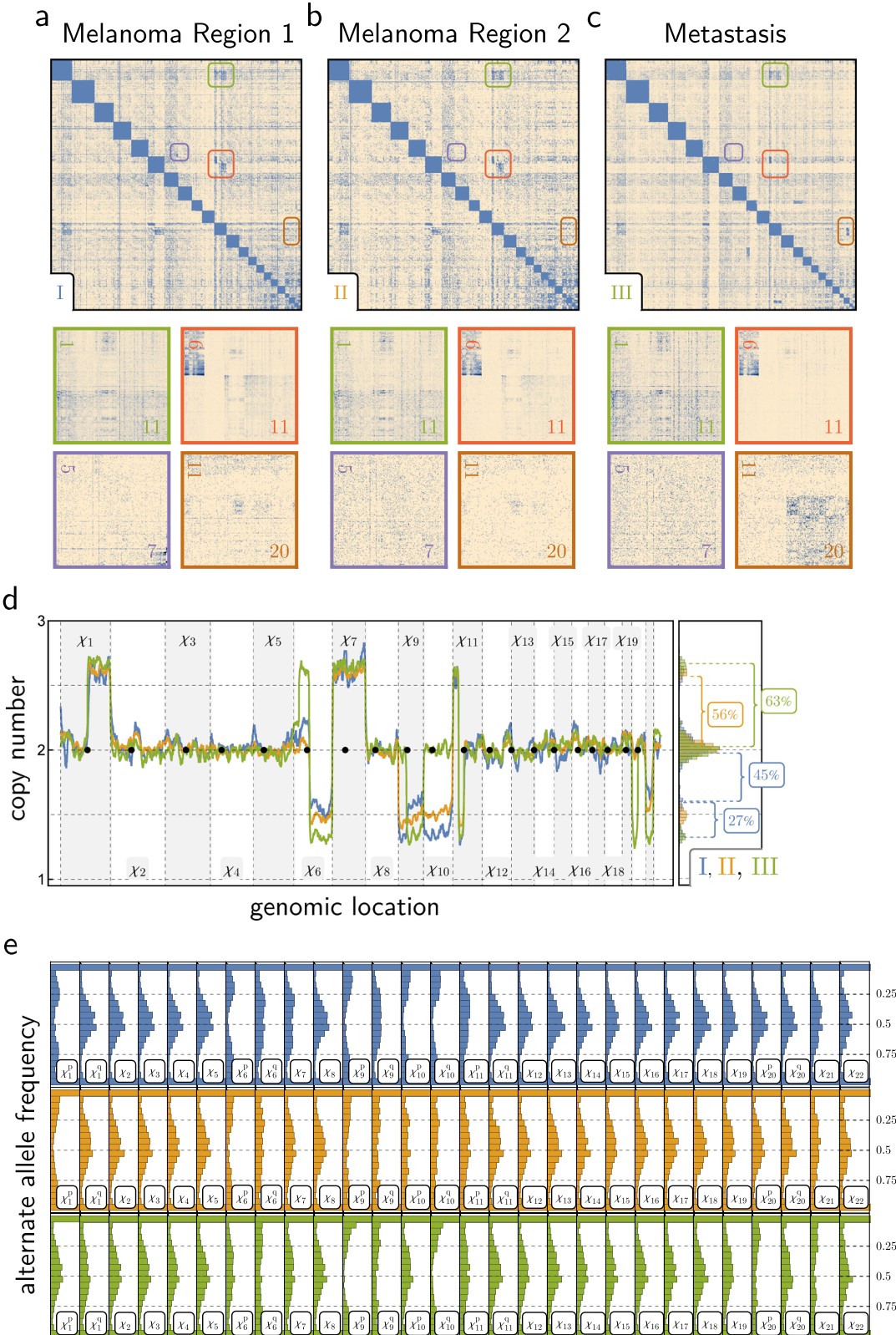

**Fig. 6 | HiDENSEC analysis of Patient 3. a–c** Hi-C maps derived from two areas of the primary melanoma (Sample 3 - I and Sample 3 - II) and a corresponding metastasis (Sample 3 - III) are shown along with insets zooming into large-scale structural variants that are observed in the three samples. **d** HiDENSEC inferred absolute copy number for the three samples along with the inferred tumor purities. Sample I in blue, II in orange, III in green. **e** Regional allele frequency spectra of common SNPs (1000 Genomes Project data[67] (Online Methods)) for I (top) II (middle), and III (bottom), colors as in (**d**), were used to track haplotypes. The allele frequency spectrum serves as an independent confirmation of the absolute copy numbers inferred in (**d**).

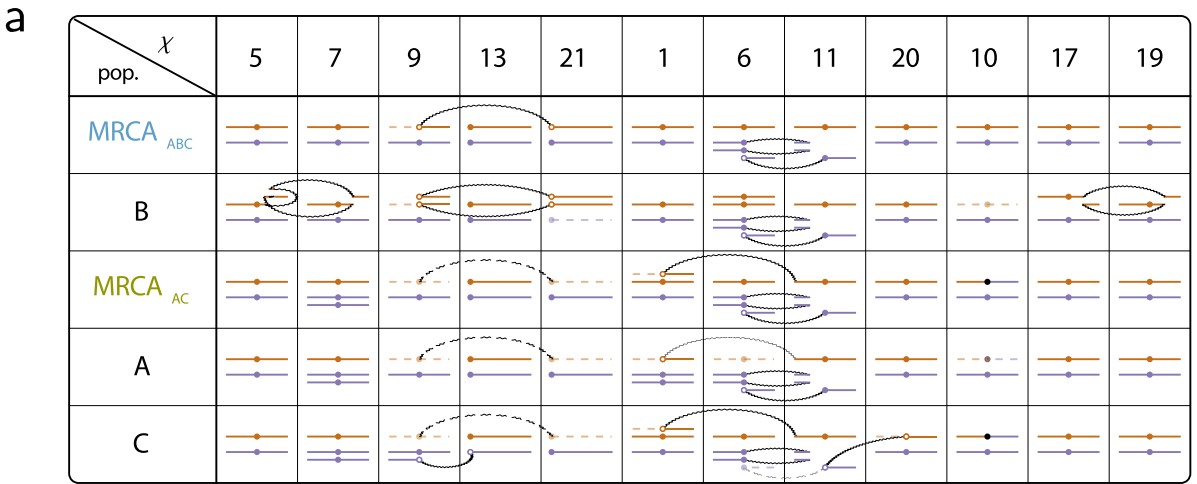

occurred, since alleles occurring at higher frequency along chromosome 7 must all lie on the mutant haplotype. Lastly, we find that genotype B carries derivative chromosomes of a reciprocal translocation between chromosome 5 and 7 (Supplementary Data 3) that heterozygously disrupts the LMBR1 gene on chromosome 7, an aberration that is absent in the A and C genotypes (Fig. 7c).

Chromosome 10, which encodes the tumor suppressor *PTEN* on its q arm, provides a more complex case. A *PTEN* Y176X mutation, which is hemizygous due to loss of the long arm of chromosome 10 carrying the wild-type allele of *PTEN*, was an early alteration found in both cell types A and B (Fig. 7c)[55]. In contrast, chromosome 10 was diploid in genotype C (Fig. 7a). Surprisingly, in genotype C the *PTEN*

**Fig. 7 | Evolution of the melanoma genome in Patient 3. a** Structural variants in the three cell populations inferred to be present in the two areas of the primary melanoma (Sample 3 - I and Sample 3 - II) and a corresponding metastasis (Sample 3 - III) (Supplementary Data 5). Notation as in Figs. 4e and 5e. Curved dashed connectors represent translocations present in the ancestor but not in the sample itself. Brown (purple) color indicates maternal (paternal) haplotype. Assignments of chromosomes to maternal (paternal) haplotypes may change across columns. **b** Inferred evolutionary changes of the three observed cancer genomes. Genetic tree of the three samples, with annotation indicating rearrangement events following standard cytogenetic nomenclature, with t(;) representing reciprocal translocations, der() indicating derivative chromosomes of such events, and plus and minus signs indicate gains and deletions. For chromosome 1, the "+1,−1" refers to gain and loss of distinct haplotypes. The patient 3 sample admits three distinct phylogenetic trees consistent with HiDENSEC's inferred copy number profile, large-scale rearrangements, and subsequent immunostaining and FISH analyses. The tree featuring the least number of independent duplicate events is depicted, with the remaining two alternatives given in Supplementary Fig. 8. *A schematic of the translocation between chromosomes 5 and 7 is depicted in Supplementary Fig. 7b, with detailed Hi-C insets of chromosomes 5, 7, 17, and 19 provided in Supplementary Fig. 9. **The precise origination of the chromosome 10q event cannot be determined from the data, and may occur anywhere prior to its current placement in the tree. **c** A phylogenetic tree derived from somatic mutations[55]. **d** Immunostaining NF1 protein in an FFPE section (single replicate) of Sample 3 - I. Circled with a black dashed line is a region of the tumor that is not immunoreactive. The inset shows a magnification of the margin between the NF1 positive and NF1 negative region. **e** Quantification of FISH analysis of FFPE section of Sample 3 - I, II and III for probes hybridizing to chromosome 6p, 6q, 11p13 and the centromere of 6. Numbers indicate signals detected per nucleus (obtained via a single replicate) and the total number of signals within the analyzed area are plotted.

Y176X mutation is homozygous with LOH along the entire q-arm, with only the p-arm remaining heterozygous (Fig. 7c). We therefore infer that the two copies of chromosome 10 in genotype C did not arise by simple chromosome duplication in the C lineage (which would have resulted in LOH along the entire chromosome), but must have involved chromosome arm exchange either in an ABC ancestor (followed by chromosome loss in A and B), or in an AC ancestor (followed by loss of the ancestral chromosome 10 in B and loss of the recombined chromosome 10 in A). Regardless of the timing of the recombination event, we infer the presence of a previously unrecognized chromosome arm exchange during progression (Fig. 7a).

The coordinated evolution of chromosomes 9 and 21 provides another example in which chromosomal rearrangements observed in Fix-C can be used to explain complex karyotypic changes. All three cancer cell genotypes have biallelic but distinct deletions of the *CDKN2A* locus on 9p encoding the tumor suppressors p14 and p16. In genotypes A and C one copy of chromosome 9 is completely lost, while genotype B retains 9q. All three genotypes carry a small deletion of *CDKN2A* on the other (blue) copy of 9p (Fig. 7a, c). Genotypes A and C are missing one copy of chromosome 21 while genome B contains a duplicated homozygous der(9q;21) chromosome arising from t(9;21)(9p13.3;21tel) (making B triploid for 9q overall) but is missing the alternate copy of chromosome 21 relative to A and C (Fig. 7a).

The most parsimonious explanation of these changes is that a t(9;21)(9p13.3;21tel) reciprocal translocation occurred in the ABC lineage, with concomitant loss of a copy of 9p (and a copy of *CDKN2A*) early in cancer evolution, presumably because the 9p fragment liberated by the t(9;21) translocation lacked a centromere (Fig. 7c). The 9q;21 fusion chromosome duplicated in the B lineage with loss of the ancestral chromosome 21 (Fig. 7a). Conversely, in the AC lineage the der(9q;21) fusion was lost, explaining mechanistically (1) the concomitant loss of the same 9q and 21 haplotypes in both A and C and (2) the loss of 9p in A, B, and C. Finally, in the metastatic lineage C a duplicated copy of the remaining intact chromosome 9 experienced a 9p::13 translocation (with loss of 9q) (Fig. 7b, c).

Chromosomes 1, 6, and 11 are the nexus of a complex series of copy number changes and rearrangements present in the three cancer cell genotypes, notably involving an early *NRAS* G12D mutation on 1p (Fig. 7c). Strikingly, as found for chromosomes 9, 21, and 13, the observed intrachromosomal copy number changes can be explained by translocations followed by consecutive mis-segregation of the resulting derivative chromosomes, rather than by direct deletion of an arm (or part of an arm) (Fig. 6b). An initial reciprocal translocation between chromosomes 6 and 11 occurred in an ancestral cell and resulted in a der(6p,11p) fusion chromosome arising from t(6;11)(6p13;11p14.3) with a 22 Mbp deletion in 11p14.3 flanking the junction. The resulting derivative chromosome underwent duplication (Fig. 7a). Subsequently, cancer genome B gained one copy of chromosome 6 while cancer genome A lost one copy. We infer that chromosome 6 loss

occurred after the establishment of the metastasis, since the metastasis – which is also hemizygous for 6q – retains the alternate homolog of 6q. In the metastasis, 6q was likely lost when the derivative 6q,11q fusion chromosome passed its 11q segment to chromosome 20q by reciprocal translocation, followed by the loss of 6q and 20p. Imbalances between the p and q arms of chromosome 6 are very common in melanoma. Gain of 6p and loss of 6q occurs ~50% of cases and can be used to diagnostically distinguish melanoma from nevi[43] using fluorescent in situ hybridization. We used this approach as an independent validation for the SCNAs changes detected by HiDENSEC (Fig. 7e).

Cancer genotypes A and C both share the loss of 1p and the fusion of the centromere of 1q with the p-telomere of chromosome 11. This coupled loss/fusion must have occurred before the most recent common ancestor of A and C (i.e., AC) and can be explained by a single event. As above, the loss of the 1p arm is presumably due to its lack of a centromere, which remained with 1q. C also retains an intact copy of the same haplotype of chromosome 1, but this is lost in A, which carries a duplicated copy of the homologous chromosome 1. The presence of an intact chromosome 1 in C implies its presence in the AC ancestor and both chromosome 1 homologs must have duplicated independently during subsequent evolution (Fig. 7a–c).

Finally, cancer genotype B harbors two balanced reciprocal translocations: (1) t(17;19)(17q11.2;19q13.32) that brings together the distal portions of 17q and 19q, and (2) t(5;7) combining the 5q and 7q arms, with in a 24 Mb deletion in the proximal 5q arm (Fig. 7a and Supplementary Fig. 9). These changes are not present in cancer genomes A and C and therefore arose in the B lineage. Notably, the junction of the translocation on chromosome 17p disrupts the *NF1* tumor suppressor gene (Supplementary Data 3). While we did not detect a mutation on the second allele of *NF1*, immunostaining for *NF1* in an FFPE section from Sample 3 - I showed a discrete NF1-negative tumor area indicating loss of NF1 protein in this region (Fig. 7d). Thus, genotype B in the primary continued to evolve after the clone forming the metastasis departed the primary, possibly under selective pressure to eliminate NF1 activity and further increase MAP kinase signaling.

## Discussion
### A new method
Solid tumors generally comprise mixtures of normal and one or more cancer cell types that typically vary throughout tumor progression, and characterizing such mixed samples requires joint estimates of cancer cell genomes along with their corresponding cell fractions. Here we present HiDENSEC, a new analytical method for investigating cancer genome evolution in patient samples using chromosome conformation capture (Hi-C). We determined the chromatin contacts in formalin fixed, paraffin embedded (FFPE) samples from three melanoma patients, using the Fix-C chromatin conformation capture protocol[29]. Using in silico and in vitro generated controls we confirm that observed Fix-C signals are linear superpositions of the signals

from normal and cancer cells weighted by cell frequency. This observation allows us to jointly estimate tumor purity and genome-wide absolute copy numbers in mixed samples, as well as to identify a large class of chromosomal rearrangements in cancer genomes, as implemented in HiDENSEC. Since the number of distinct genotypes in a mixture is underdetermined from bulk data, HiDENSEC finds the most parsimonious explanation that is statistically compatible with the data. We find that our melanoma data can be explained by mixtures of normal cells with one or two cancer genotypes, although we cannot rule out additional low-frequency clonal sub-populations, as observed in other contexts[48–50]. The sensitivity of our method allows us to detect rearrangements that occur in melanoma development and to define the genetic changes that occurred specifically in minor subpopulations of melanoma cells. For example, in one deeply sequenced sample (3 - I) we find a subpopulation with frequency of 12%. Hi-C data also provides information about allele frequencies, which can be used to identify copy-number neutral losses of heterozygosity or to identify the more common haplotype in triploid situations.

Since chromatin contacts probed by Hi-C typically extend over hundreds to thousands of kilobases, the method allows us to capture large-scale (>1 Mb) rearrangements. In contrast, conventional short-read sequencing approaches rely on mapping read pairs across rearrangement junctions and thus may have lower sensitivity for discovering copy-number neutral rearrangements and may fail to map events with breakpoint in repetitive sequences. For detecting smaller-scale (<1 Mb) rearrangements short-read sequencing-based methods complement Hi-C based approaches described here. Our benchmarking experiments for the detection of chromosomal rearrangements/aneuploidies in Hi-C data sets showed that HiDENSEC was more sensitive and accurate in identifying such karyotypic alterations than the current gold standard; indeed, we find that Hi-C clearly reveals rearrangements that are not described in standard structural variation benchmarks.

A key advantage of the analysis of fixed tissue over fresh tumor samples is that material collected over the course of a patient's disease progression can be analyzed retrospectively to explore the temporal evolution of cancer. Fix-C can be carried out from small amounts of material obtained from thin sections or micro-dissected samples based on morphology or pathology. The sensitivity of our method allows us to explore the spatial differences across such samples containing small amounts of DNA.

## Biology of melanoma progression

Chromosome arm aneuploidies are widespread in cancers, and specific co-occurring chromosome arm deletions have been linked to prognosis and drug response[8,9]. Studies focused solely on copy number alterations, however, cannot fully characterize the mechanism of correlated loss. The combination of Fix-C and HiDENSEC can detect these karyotypic changes. We find that correlated arm losses are mediated by reciprocal translocations followed by the loss of one derivative (bi-armed) chromosome; in one case an arm is passed from chromosome to chromosome by sequential translocations before deletion. While mechanistic analyses of chromosome-arm co-deletion have been reported using cytogenetic methods (e.g., t(1;19) mediating the combined deletion of 1p and 19q in oligodendroglioma[57]), these traditional approaches require cell culture from fresh samples. Here we find that HIDENSEC can be used to trace complex translocation and loss events during cancer evolution using fixed samples. The connection between chromosome-arm aneuploidy and drug response[8,9] suggests that Hi-C based approaches may contribute to precision oncology.

The long-range information inherent in chromatin capture data is particularly useful for identifying rearrangements whose breakpoints lie in repetitive regions. In particular, the centromeric breakpoints or telomere fusions repeatedly found in our melanoma analyses would be difficult or impossible to detect by conventional sequencing. Specifically, in the three patients' samples analyzed, HiDENSEC was able to annotate a total of three fusions involving telomeres (one of which is present in all three samples of patient 3), five chromosome arm exchanges with breakpoints in centromeric regions, and several copy number neutral reciprocal or complex translocations. Our approach therefore provides an integrated picture of cancer heterogeneity and karyotype evolution in melanoma. A common process in our melanoma progression cases is whole chromosome arm rearrangement followed by loss and/or copy number change by mis-segregation of derivative chromosomes. The most complex changes were found in the acral melanoma from patient 3. For example, a chromosome 6 to 11 translocation was followed by a subsequent translocation, so that the 11q of the derivative chromosome became fused to 20q with the concurrent loss of 20p.

By sampling and analyzing several progression stages of three different melanomas we infer the evolution of genome organization across multiple subpopulations of cancer cells. Comparing these subpopulations and applying the principle of maximum parsimony along with the known temporal relationships among the samples, we can infer unsampled intermediates and possibly transient states in cancer progression (Fig. 7a). These comparisons took advantage of HiDENSEC's ability to analyze mixed samples and estimate tumor cell fraction. In patient 3 we find that of the three subpopulations detected, melanoma genotype A is more closely related to the metastatic genotype C subpopulation, and that melanoma genotype B diverged prior to the A-C divergence. This, in turn, allows us to characterize the changes that occurred on this cellular phylogeny. We find that the most recent common ancestor of genotype A and genotype C is linked to large structural events that resulted in a gene conversion event of large parts of the q arm of chromosome 10 (Fig. 7b).

Our analysis of two regions of the melanoma from patient 3 highlights the karyotypic heterogeneity in this tumor. In Sample 3 - I, an area of primary melanoma, two genetic subclones were identified. Previous analysis of this patient identified several consecutive mutations that lead to upregulated MAPK signaling by different mutations including, including copy number gain of *BRAF* in the AC precursor and loss of heterozygosity for the *NRAS* G12D mutation in melanoma genotype A (Fig. 7c)[55]. The identification of structural rearrangement that disrupts the *NF1* locus in subpopulation B illustrates that HiDENSEC can uncover novel large-scale chromosomal rearrangements and aneuploidies that drive cancer cell evolution, even when only present in small cancer cell populations. While *NRAS* mutant cancers typically do not have *NF1* or *BRAF* mutations, the G12D mutation likely still has some residual GTPase activity, explaining how *NF1* loss and *BRAF* mutation provides a selective advantage for this branch of the melanoma evolution in patient 3.

Together our analyses of samples from three cancer patients demonstrates that HIDENSEC analysis of FIX-C data can characterize cancer cell genome evolution from early stages of cancer development using microdissected tissue. Notably, this approach allows us to deconvolve heterogeneous mixtures of cancer cells with distinct genotypes and follow the genomic changes through time by analyzing samples through cancer progressions. Applying this approach at a larger scale to investigate will significantly enhance our understanding of cancer cell genome evolution by revealing common patterns of chromosomal change that can be used both for diagnostic purposes and to further decipher the underlying causal genetic changes during cancer progression.

## Methods

### Source and characterization of melanoma samples

Archival formalin-fixed, paraffin-embedded (FFPE) melanoma samples were retrieved from the archives of the University of California San Francisco Dermatopathology service, under an IRB approved protocol

(11-07951). Routinely stained sections were evaluated and tumor areas were marked by a dermatopathologist. Tumor-bearing areas for patient 1 and 2 were microdissected from 10 μm-thick unstained sections, using HE-stained sections as guidance. Samples for patient 3 were analyzed without microdissection. FISH was performed with locus-specific probes for chromosomes 6p (*RREB1*), 6q (*MYB*), 11q13 (*CCND1*), and 6 centromere as previously described[58].

### Fix-C methodology and sequencing

FFPE sections were processed using Fix-C® kits from Dovetail Genomics. The sample preparation and Fix-C protocol were performed as described previously[59]. Briefly, paraffin embedded tissue was dissolved in xylene followed by centrifugation. The tissue sample was hydrated with a series of ethanol washes (100%, 70%, 20%) and water followed by centrifugation. The tissue sample was digested with proteinase K at 37 °C for 1 h. The digested sample was centrifuged and the supernatant was saved to capture chromatin on beads. The chromatin was digested with DpnII restriction enzyme at 37 °C for 1 h followed by wash. The digested ends were repaired and subjected to proximity ligation at 16 °C for 1 h. Post ligation sample was crosslink reversed and DNA was purified on AMPure XP beads. The purified DNA was sheared and end-repaired for Illumina adapter ligation. The proximity-ligated DNA is enriched with capture on streptavidin beads. The captured DNA is then PCR amplified on beads for 13 cycles, purified using AMPure XP beads, quantified, and sequenced.

### Cell culture and formalin fixation of cell line mixtures embedded into paraffin blocks

Suspensions of a normal human cell line (GM12878) were cultured in RPMI-1640 medium [ATCC] supplemented with 15% FB Essence [Seradigm] and 100 U/mL Penicillin-Streptomycin [Gibco]. Mouse embryo fibroblasts (MEFs) were cultured in DMEM [Gibco] supplemented with 15% FB Essence and 100 U/mL Penicillin-Streptomycin. The human and mouse cell lines were dissociated by Trypsin-EDTA (0.25%) [Gibco] for single cell suspension, then quantified by Trypan blue staining with a Countess cell counter [Invitrogen]. 200 μL of 2% agarose in PBS solution was pipetted into a 1.7 mL microfuge tube and allowed to solidify. Equal numbers human and mouse cells (15 million total) were mixed and pelleted in a 15 mL conical tube, then resuspended in a small volume of neutral-buffered 10% formalin, then finally re-pelleted in the microfuge tube with agarose plug. Supernatant was then aspirated and fresh neutral-buffered 10% formalin was gently pipetted onto the cell pellet. The microcentrifuge tube was then placed in buffered formalin at room temperature for 24 h. The bottom of the microcentrifuge tube was then cut off with a razor blade and the plug gently extruded into a tissue cassette immersed in PBS using a pipette tip. The tissue cassette with mixed cell line plug was then embedded in paraffin and sectioned using standard protocols[60].

### Creation of in vitro normal–cancer mixtures

Adherent human cancer cells (HCC1187) were cultured in RPMI-1640 medium supplemented with 10% FB Essence and 100 U/mL Penicillin-Streptomycin. Human wild-type (GM12878) and HCC1187 cancer cells were dissociated by Trypsin-EDTA and mixed in ratios of 1:1, 2:1, and 4:1 WT:cancer cells before pelleting, fixation, paraffin-embedding, and sectioning as described above.

### Creation of in silico normal-cancer mixtures

We obtained in situ Hi-C data for the karyotypically normal cancer cell line, GM12878 from the 4D Nucleome Data Portal[61] (4DNFIYECESRC) and also generated Hi-C data for the cancer cell line HCC1187C. To create in silico mixtures of the karyotypically normal and cancer cell line, we then normalized both of them with respect to total coverage and computed convex combinations of the resulting Hi-C matrices at 50Kb resolution, to achieve any given tumor purity level.

### Allele frequency spectrum

To infer haplotype copy number, we computed the regional frequency of nominally inherited variants in 500 kb windows. A single copy of each haplotype would then have a peak at 50% frequency; two copies of one haplotype and one copy of the other would appear as peaks at 33% and 67%; loss of heterozygosity would appear as peaks at 0 and 100%. Since we did not have matched patient normal samples, we considered variants as nominally inherited if they occurred with alternate allele frequency between 40% and 60% in the 1000 Genomes Project. Note that this allele frequency spectrum (as shown, e.g., in Fig. 6e) does not include somatic mutations, which are considered separately.

### Running HiNT, hic_breakfinder, EagleC, and HiCTrans

HiNT[36] was run with version: 2.2.7. hic_breakfinder[38] was built from the master branch source downloaded from github, https://github.com/dixonlab/hic_breakfinder, using commit 30a0dcc6d01859797d7c263df7335fd2f52df7b8 (last updated in 2018). For hic_breakfinder the inter and intra chromosomal break files were provided by the Dixon lab as detailed on the github site. NextFlow pipelines to run both HiNT and hic_breakfinder can be found on the HiDENSEC github. EagleC[37] was downloaded from https://github.com/XiaoTaoWang/EagleC and run with --prob-cutoff set to 0 so as to allow for calling of all possible variants. We were unable to install HiCTrans[62] despite interactions with the authors due to deprecated dependencies of the hashmap R package.

### HiDENSEC pipeline

Hi-C paired-end (PE150) sequencing reads were aligned to the hg38 reference genome using bwa[63] and then converted to Hi-C maps using Juicer[64] and visualized in Juicebox[65]. These were then processed with the HiDENSEC pipeline which comprises custom pre-processing scripts as well as a Mathematica Notebook reproducing all presented results. The HiDENSEC pipeline is available at https://github.com/sanjitsbatra/HiDENSEC. HiDENSEC aims to interpret the Hi-C contact map of a cancer sample as a mixture of cells with distinct genomic types. Each genome has a discrete set of copy number changes and rearrangements relative to the diploid genome, and occurs in a fraction of the cells in the sample. Both the copy numbers, rearrangements, and cell fractions will be inferred from the Hi-C dataset, typically including one normal or wild-type genome and one or two aberrant cancer genomes. In order to arrive at the copy number profiles, rearrangements, and the tumor purity of each genome, HiDENSEC proceeds in broadly three steps: (i) covariate correction, (ii) joint inference of absolute copy number profile and tumor purity, (iii) detection of large-scale structural variants. (ii) and (iii) occur partially in tandem in order to facilitate sharing of information that may improve statistical inference. Despite the large number of cells contributing to any single Hi-C experiment, read counts in general do not tend to follow parametric distributions typically associated with increasing sample sizes, and so HiDENSEC remains fully nonparametric throughout all these steps. These steps typically result in unique genome configurations returned by HiDENSEC. Occasionally however, they may lead to multiple estimates that are consistent with the Hi-C data, in which case additional analysis of allele frequencies, immunostaining and FISH are used for disambiguation.

### Covariate correction

It is known that a variety of biological and experimental factors affect relative Hi-C read counts, and thus correcting their impact is essential for both unbiased and stable inference. The most prominent such factors include GC (guanine plus cytosine) content, mappability, cut-site density, and compartment structure. The first three of these covariates were also modeled for the human reference genome, hg38, by HiNT[36]. Compartment structure was obtained for the karyotypically

diploid GM12878 cell line[46]. Concretely, HiDENSEC models observed read counts falling into a bin of length $w$ around a site $i$ as

$$reads_i \approx (absolutecopynumber)_i \\ \cdot correction\,(GC_i, mappability_i, cut\,sites_i, compartment_i) \quad (1)$$

where the corrector function is a simple compartment-specific linear model:

$$correction\,(GC_i, mappability_i, cut\,sites_i, compartment_i) \\ = \sum_{c \in compartments_i} 1_c \times (\beta c,1 \cdot GC_i + \beta c,2 \cdot mappability_i + \beta c,3 \cdot cut\,sites_i) \quad (2)$$

The linear models are chosen to match observed trends in diploid reference genomes, and reliably account for ≈80% of their variability. The coefficients $\beta c,\cdot$ generally depend on the precise experimental details (e.g., whether Hi-C or Fix-C protocols were used), and so we recommend using reference maps obtained through the same experimental protocol as the map of interest for performing the covariate correction. In the absence of such a reference map, HiDENSEC defaults to performing the correction (1) and (2) internally within the map of interest on only those genomic sites that are likely to be diploid (see section below). For the samples presented in the main results, protocol-matching reference Hi-C maps are available, and so have been used throughout. We note that the correction procedure in the form given by (1) and (2) only applies to the diagonal entries of the binned Hi-C matrix—as information about copy number profiles is almost entirely contained therein—and leaves off-diagonal components unchanged. Since off-diagonal read counts indicating contacts between (binned) site $i$ and $j$ are primarily used for detecting fusion events, their precise magnitude beyond a broad distinction of large and small (corresponding to the presence or absence of fusion events) is substantially less informative (see Supplementary Fig. 4). Thus, even though it is straightforward to extend (1) and (2) to correct off-diagonal read counts by regressing against paired covariates, such extension likely does not increase accuracy, and so is not part of HiDENSEC.

## Inference of copy numbers and mixture proportions

Our goal is to estimate both the absolute copy number profiles (i.e., integer-valued local ploidy along the genome) and mixture proportion for each of the constituent genomic types. This is, however, an ill-defined problem without additional constraints. First, the Hi-C contact map cannot distinguish between uniformly diploid and uniformly triploid genomes (although this can be done by measuring allele frequencies which will differ in these two cases). Second, we cannot distinguish between a 50–50 mixture of a wild-type genome with a cancer genome bearing a triploid chromosome 1, vs, a 75–25 mixture of a wild type with a cancer genome bearing a tetraploid chromosome 1. Third, the number of subpopulations is not identifiable. E.g., the Hi-C map of a single subpopulation with many translocations is indistinguishable from a map involving many subpopulations (one for each translocation), as long as each subpopulation's mixture proportions are similar. All of these kinds of ambiguities arise even in the absence of noise and make the problem under-determined without additional assumptions. To address these unidentifiabilities, HiDENSEC operates under three corresponding assumptions:

1. The most common absolute copy number (of the mixture) is known. Knowledge of this copy number mode allows for appropriate rescaling of the Hi-C matrix correcting for the unknown constant $C$. Other known statistics of the absolute copy number profile may be used; however, the mode is particularly appealing since it most often will equal 2, and as it is particularly reliable for estimating $C$.

2. Absolute copy numbers are as close to diploid as consistent with the data. That is, HiDENSEC returns the biologically most parsimonious estimate.

3. The data was generated by the smallest number of subpopulations consistent with it. That is, HiDENSEC again is guided by parsimony.

Given these assumptions, HiDENSEC appropriately centers the (covariate corrected) read counts by their largest mode, and infers absolute copy number profiles and mixture proportions jointly, one cell population at a time, by scanning along the genome in overlapping windows of length $w$, typically taken to be 50 or 100 kb, and identifying for each such window and a fixed choice of mixture proportion the copy number value that minimizes a suitably designed metric between predicted copy number and observed Hi-C intensities. A resulting global discrepancy metric is then minimized over all choices of mixture proportions, yielding both an overall mixture proportion estimate as well as local absolute copy number inferences. This estimated copy number profile is then subtracted from the read count data, and the entire procedure repeated in order to detect any potential further subpopulations contributing to the Hi-C matrix.

It can be shown (see Supplementary Note 1) that the inference scheme described above provably recovers the correct mixture proportions and absolute copy number profiles in the limit of noiseless data and comparatively few distinct cell populations, or in the case of cell populations, whose mixture proportions and copy number profiles satisfy certain monotonicity properties (broadly, if more abundant cell populations exhibit sufficiently many copy number changes that are not shared by the less abundance cell populations, then inference is accurate). The latter constraint is not surprising, since the general inference task tackled by HiDENSEC is NP-complete in the number of cell populations, while the algorithm described above scales linearly in them. In order to relax the former constraint and accommodate noisy data, HiDENSEC performs a number of additional refinement, model selection and hypothesis testing steps that correct for any copy number changes that may be called purely as a result of random fluctuations or whose precise location may be shifted as a result thereof. In the process, each change point is assigned an interpretable confidence score that indicates to what extent it is likely to reflect actual biological signal, as opposed to being the outcome of noise. After undergoing another round of refinement using detected off-diagonal events (see section below), these estimates are then returned to the user for interpretation.

Likewise, HiDENSEC refines the initial mixture proportion estimate based on similar principles, and additionally equips them with 95% confidence intervals that reflect their associated uncertainty. There are two primary sources that contribute to this estimation uncertainty: Stochastic fluctuations in read counts and uncaptured biological or experimental covariates. While the former is typically well-addressed by classical non-parametric tools like the bootstrap, the latter is more delicate and possibly instance-specific, prompting HiDENSEC to employ bootstrap ideas combined with structured subsampling that integrate information within and across individual stretches of copy-number changes. The resulting confidence intervals are conservative, yet not overly so; see Supplementary Fig. 3 for details.

## Inference of large-scale structural variants

Large-scale genomic rearrangements typically result in off-diagonal chromosome x chromosome submatrices of the Hi-C contact maps that are structured in either of the six patterns described in Supplementary Fig. 14. The first four patterns are denoted as the set $P_1$, while the latter two structures are evidence of reciprocal exchanges and are denoted as $P_2$ in Section 4 of Supplementary Note 1. The events in $P_2$, due to their lack of rotational and translational symmetry compared to

non-reciprocal events, are generally easier to identify. On the other hand the events in $P_1$ may indicate not only large scale rearrangements, but can also result from DNA geometry, compartment structure, or simply stochastic fluctuations, and are thus found abundantly throughout Hi-C maps. Distinguishing fusion events from these biological and experimental confounding can thus be difficult, especially when faced with particularly noisy data. Moreover, once stochastic fluctuations become sufficiently strong, the intensity gradients within each sub-matrix may wash out, effectively rendering all of them rotationally and translationally equivalent. To address these sources of uncertainty, HiDENSEC resorts to two corrections:

1. HiDENSEC only aims to detect non-reciprocal fusion events of type-1 as described in the main text. Due to their effect on local copy numbers, events in $P_1$, corresponding to type-1, allow HiDENSEC to rely on its previously inferred copy number profile to aid in their detection. More concretely, by default HiDENSEC will only consider off-diagonal sub-matrices anchored at coordinates associated with copy number changes deemed significant by the previously outlined analysis. Switching to non-default behavior and scanning points along the whole genome is possible, but care should be taken in interpretation, as confounding by above-mentioned biological and experimental covariates may be present. Additionally, restricting HiDENSEC's search to copy number change points drastically reduces its run-time, with a typical analysis completed in less than twenty minutes on a typical laptop.

2. Experimental and biological confounders tend to affect rows and columns more globally. Biological confounders like compartment structure generally elevate read counts of interactions between the region of interest and all other sites in the genome, leading to entire rows and columns in the Hi-C matrix that are enriched. All summary statistics computed by HiDENSEC are thus calibrated by comparing their value at the site-pair of interest against their empirical distribution across the associated row and column.

With these two corrections at hand, HiDENSEC considers two summary statistics that essentially measure the extent to which (a) intensities tend to accumulate in only one of the four quadrants of each sub-matrix, and (b) large- and small-intensity regions are separated by clear boundaries or edges. Under suitable null hypotheses on the Hi-C read count distribution, and as these two summary statistics are normalized against their row and column histograms, the corresponding $p$ values are readily combined, yielding properly controlled aggregate $p$ values based on which HiDENSEC calls significance.

As sub-matrix patterns in $P_2$ are typically not tied to changes in copy number profiles, detecting potential candidates requires a more global search. Because such potential candidates are generally distinguished by dense patches of large intensities, HiDENSEC does so by effectively enumerating the largest connected components of a suitably obtained graph that respects the geometric structure of the Hi-C matrix, and inspecting its point of largest intensity, or focal point. Once these candidates are determined, a number of summary statistics aimed at capturing (a) concentration and sharpness properties as with events in $P_1$, (b) enrichment near a central focal point, and (c) the presence of a gradual intensity decrease away from the focal point, are computed, and their calibration under suitable null hypotheses again verified (see Supplementary Note 1 for details).

### Reporting summary
Further information on research design is available in the Nature Portfolio Reporting Summary linked to this article.

### Data availability
The Hi-C and capture sequencing (UCSF500 or Exome sequencing) data generated in this study have been deposited in the dbGap database under accession code phs001550.v3.p1. Additionally, the Hi-C data generated in this study for the in vitro mixtures have been deposited in the SRA database under accession code PRJNA849975. Source data for figures is available on Zenodo [https://doi.org/10.5281/zenodo.8313343].

### Code availability
The code to infer absolute copy number, ploidy, tumor purity, and large-scale rearrangements from a Hi-C data is provided at https://github.com/songlab-cal/HiDENSEC.

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

## Acknowledgements

Y.S.S., D.S.R., and D.H. are Chan Zuckerberg Biohub Investigators. D.H. is a Pew-Stewart Scholar for Cancer Research supported by the Pew Charitable Trusts and the Alexander and Margaret Stewart Trust. The work in the Hockemeyer laboratory was supported by the Siebel Stem Cell Institute and D.O.D. (W81XWH-19-1-0586) and by a Research Scholar Grant from the American Cancer Society (133396-RSG-19-029-01-DMC). The research in the Song lab was supported in part by an NIH grant R35-GM134922. D.S.R. is grateful for support from the Marthella Foskett Brown Chair in Biological Sciences. I.Y., H.S., B.C.B., and D.H. are supported by a Team Science Awards of the Melanoma Research Alliance. B.C.B. is supported by NIH grant 1R35CA220481.

## Author contributions

D.H., D.S.R., and B.C.B. conceived and designed the study. M.B., T.K.T., D.H., I.Y., H.S., and B.C.B. performed data collection, experimental data generation, and experimental analyses. D.D.E.-P., S.S.B., Y.S.S., H.S., J.D., D.S.R., and D.H. analyzed experimental data and interpreted results. All authors contributed in preparing the manuscript, reviewed the results, and approved the final version of the manuscript.

## Competing interests

D.S.R. is a paid consultant and equity holder in Dovetail Genomics. J.D. and M.B. are employees of Dovetail Genomics. The other authors declare no competing interests.
