## [Peer Review File · Nature Communications]

Tracing cancer evolution and heterogeneity using Hi-CREVIEWER COMMENTS

Reviewer #1 (Remarks to the Author):

This study from Erdmann-Pham and Batra describes a new computational method, HiDENSEC, for detection of rearrangements and copy number changes in patient tumor samples analyzed with Hi-C/Fix-C data. The study makes use of several novel computational strategies for identifying rearrangements and copy number changes in Hi-C data, and applied this tool to three patient samples with multiple sections to study the evolution of the karyotypes in these samples. On the whole, detection of rearrangements and copy number changes using Hi-C data has been demonstrated by several groups previously, and at least one study which the authors cite have performed such analysis on FFPE samples. Further, while it is interesting that they can find events disrupting NF1, CDKN2A, and PTEN, these are all pretty well known events in human cancers, and I think they are really underpowered to make meaningful conclusions regarding biological novelty from the 3 patient samples. The real technical novelty to me seems to be in their computational tool, for example in the estimation of tumor purity for Hi-C data. However, tools for estimating tumor purity have been established for WGS derived data, so the advancement introduced here is pretty modest. They also utilized several novel strategies to detect copy number changes and rearrangements. However, I find that they have not really extensively benchmarked their method against alternative approaches for detection rearrangements/CNVs, so the improvement in performance of their approach I don't think has been sufficiently demonstrated. While I find aspects of this study interesting, it seems somewhat preliminary in terms of novel biological insights or in the demonstration of improved computational ability for CNV/SV detection. I have divided my comments into major and minor comments below.

Major comments:

The authors need to improve their benchmarking against known gold standard and against other methods for detecting rearrangements and copy number changes. They compare with HiINT, but there are other methods for detecting rearrangements in Hi-C data (HiCtrans, EagleC, hic-breakfinder) that are untested. They also do not perform comparisons with

other methods for detection of CNVs (though they cite several tools that are used to do so).

The methods for assigning “gold standard” SVs are pretty ad hoc. There are plenty of samples with high quality reference datasets in terms of whole genome sequencing, RNA-seq, or other methods they could use as an unbiased “gold standard.”

Hi-C also has a known bias for detecting shorter rearrangements that are more readily detected by WGS, but there isn’t a comparison of the ability of their method to

They test the ability of the HiDENSEC method for recall, but not necessarily for false positives. The improvements seen in HiDENSEC (Fig. 3b) could be due to it being more “aggressive” in making calls. They should estimate the fraction of SV calls made by HiDENSEC that are “true” from gold standard datasets.

Minor comments:

In the legend for Figure 1, they say that “FFPE samples may be microdissected.” This is pretty vague. Do they perform microdissection? If so, say it and explain how. If not, this should be deleted (or moved to the discussion where speculative statements are more appropriate).

I think it is a bit of an overstatement to say that HiNT is the gold-standard for Hi-C analysis. Other newer methods, such as EagleC, have also been introduced that have very good performance.

The authors state (line 124) that “A novel feature of our method is correcting the HiC signal for covariates including chromatin compartment and GC content, which allows accurate determination of copy number and tumor purity, and higher confidence detection of interchromosomal rearrangements.” Correction for GC content has been described in numerous previous studies analyzing biases in Hi-C data (PMID: 22001755, PMID 23023982), so this isn’t a particularly novel idea.

The authors state (139) “The application of high-throughput chromatin conformation capture methods to FFPE samples (Fix-C) opens new possibilities for studying chromosome rearrangements in solid cancers.” While I think that the Fix-C approach to studying FFPE samples is pretty cool, it isn’t strictly necessary for studying solid tumors. Several previous studies have used solid tumor samples to generate Hi-C data previously (PMID 32841603, 28655341), so this is a bit of an overstatement.

The authors state (line 225): “We therefore only used the presence/absence of these off-diagonal signals to detect large structural variants, and inferred absolute copy numbers and tumor purity based on the on-diagonal intensity analysis as described above (Supplementary Note 1).” My question related to this point is for balanced translocations. Can they detect absolute copy numbers under such circumstances that do not involve copy number changes? From the methods I get the impression that they don’t do this (or at least from Supp. Fig. 4 believe it is not accurate), but to the best of my knowledge estimating absolute copy of balanced rearrangements is something that is not possible in WGS data, and this study would represent the first time I am aware of that such signals might be estimated.

The Figure legend for Fig. 3a is a little confusing, particularly the meaning of the dashed versus solid lines. Which represents type 1 vs type 2 vs. all? Do they show both type 1 and type 2?

They seem to have included a note to “[cite everyone]” in the introduction (line 91).

Reviewer #2 (Remarks to the Author):

This manuscript from Erdmann-Pham et al presents the development of HiDENSEC, a new tool that uses Hi-C data (Fix-C chromatin conformation capture protocol) to investigate genome evolution. Key strengths of HiDENSEC are an ability to extract useful information from samples that have been formalin fixed, paraffin embedded, identify chromosome rearrangements that are copy number altering and copy number neutral, maintain sensitivity even when cancer cell populations are diluted by normal cells, estimate tumor purity and absolute copy number, and determine allele frequencies. Other key strengths are

an ability to map rearrangements at repetitive sequences like centromeres and telomeres. Benchmarking analyses indicate that HiDENSEC compares favorably to a similar method named HiNT. The authors use HiDENSEC to investigate chromosome rearrangement architectures across three melanomas and associated nevi. Interestingly, HiDENSEC powers the authors to identify telomere fusions in melanoma genomes, a difficult feat to accomplish using traditional short-read sequencing. Overall, the method development is clearly articulated and the experiments are rigorously performed. HiDENSEC represents an important new tool for researchers seeking to characterize cancer genomes across time in heterogeneous samples. There are no major surprises or insights gained into melanoma genome evolution described here, however this is a minimal limitation in an article focused on method development.

The authors compare HiDENSEC against HiNT, but do not extensively characterize HiDENSEC against more commonly used gold standards like paired-end sequencing approaches. A key potential advantage of short read sequencing is the ability to call base substitutions alongside chromosome rearrangements. I do not think further experimentation is necessary but it would be helpful to non-specialists to make room for these comparisons in the discussion section.

One minor note is that I spotted several typographical errors throughout the text. For example: 'Cite everyone' typo on line 91; Line 94 'decay' typographical error.

Reviewer #3 (Remarks to the Author):

In this study, the authors present an approach that exploits Hi-C data generated from FFPE tumor samples to infer tumor purity, absolute copy number status, and complex rearrangements. The idea is overall interesting and well implemented with supporting results, although it feels as if several conclusions were rushed and substantial additional evidence should be shown to support these conclusions. Beyond my main comments listed below, the feeling of incompleteness is further conveyed by multiple imprecisions throughout the text, as if this was a draft and not yet a definitive version of the manuscript (see lacking or incorrect definitions, missing and misplaced references to figures, inconsistent annotations,

etc.). I reported some of them at the end of this report, but I would strongly encourage the authors to carefully revise the text in its entirety.

Major comments:

1) The potential issue of tumor clonal heterogeneity is briefly touched upon by the authors but quickly dismissed saying that "it is typically sufficient to include one or sometimes two cancer genotypes along with the diploid reference genome" (from Supplementary Note 1). This is a bold statement and not necessarily supported by the literature where tumors exhibiting more than 2 clonal subpopulations have been reported in multiple contexts (see for example PMID:31209394, PMID:28445469, PMID:29656895 and many more).

Are there truly at most 2 clones in these samples or you cannot detect more than 1 or 2 given the resolution of the experiment? Indeed, the Hi-C experiments used in the manuscript seem to have an overall low number of reads: sample 2 I and II are both below 50 mln, which is very low considering these are admixes of normal and cancer cells, and only sample 3 I and II have more than 150 mln; coincidentally this is the sample where more than 1 clone is detected. The authors should introduce a control e.g. by performing whole genome sequencing of these samples and comparing the estimated clonality and/or by generating synthetic samples with more than 2 populations and testing the limit of detectability of their approach (e.g., combining 2,3,4,5,6 cell populations at different proportions). The latter should be doable considering the large amount of Hi-C datasets available for cancer cell lines.

2) The authors have only compared their approach with HiNT, claiming this is the gold standard for these types of analysis from Hi-C data. While I can trust that, the truth is that for these analyses (purity, copy number status, and, especially, structural variants), the true gold standard is still whole genome sequencing. A comparison of their results with those obtained by whole genome sequencing data is warranted. The authors may search and use available datasets where both information are accessible (these exist again for several cancer cell lines) and compare the copy number and structural variant calling ability.

3) Another undiscussed issue concerns compartment differences among different cell lines. First of all, although never explicitly mentioned, based on supplementary information I'm guessing that the authors are using the compartment calls generated in (Rao et al. 2014 - PMID:25497547) for the GM12878 cell line. Is this correct?

(I assumed so because in the supplementary note the authors mentioned the use of 6 sub-compartments, which is the number proposed in that study, although the labeling is inconsistent with that proposed in that study, it should be A1, A2, B1, B2, B3, and B4. Unless of course this classification comes from somewhere else.)

The problem is that different cell types can have very different compartments and while on a global scale the correlation will still be high, I wonder whether local differences at specific loci may affect copy number calls at that specific loci. This should be tested.

This is of course a complex issue because in the admix of multiple cell types (e.g., normal and cancer)

you will have different compartments being admixed from highly heterogeneous immune, stromal, and cancer cells.

A possible solution could be regressing only with respect to loci that have been shown have highly consistent compartment assignments across multiple cell types (e.g., as shown in PMID:33972523 - data should also be available). The authors should test this possibility, even if in their samples it won't change the conclusions, ignoring compartment differences is a risk.

4) This might be a minor point but I wonder why the authors do not try to infer the "purity" of translocations/re-arrangements. They say that doing this at off-diagonal entries is difficult and subject to biases. But bins around the breakpoint (or even where the breakpoint is) are only seemingly 'off-diagonal' but they are actually on the diagonal of the chimeric chromosome, so it should be possible to apply the same approach. This could be interesting if it allows to estimate the fraction of cancer cells exhibiting a given variant.

5) The authors introduced multiple tumor phylogenies and genotype reconstructions based on what is stated to be a maximum parsimony approach, but it looks like this was all done

by manual curation. Is that so? The authors should elaborate more on the procedure that was followed and if these results come from their intuition and common sense, it should be said explicitly (and possible alternative approaches should be tested).

6) Lastly, the discussion currently reads more as a summary of the results rather than a true discussion. Can the authors elaborate more on the current limitations, possible improvements, and broader applicability (e.g. compared to WGS) of their approach? I would further discussed for example the limited resolution of the Hi-C datasets, is this intrinsic of working with FFPE samples or could it be improved with further sequencing?

Minor imprecision in the text (here I indicated a few but there are likely more)

- line89: Hi-C actually stands for 'high-throughput chromosome conformation capture'

- line91: "[cite everyone]" I believe there is a missing citation here?

- line94: "decay slowly" - I would not define an exponential decay as 'slowly', what would be fast then?

- line107: "the current gold-standard for analysis of Hi-C in data" what does that mean? I believe something is missing in this sentence

- Figure 1 is not referenced in the text

- line 248: from the description there are 8 samples not 9

- suppl. fig 7 only shows schematics of the translocations for patient 2, and it points to suppl. fig 9, but here the legend refers to patient 3 - which is it?

- line 319 and paragraph below: figure references to Fig 6 and Fig 7 seem misplaced and/or missing

- in the online methods, covariate correction equation has black boxes

REVIEWER COMMENTS

We would like to thank the reviewers for their insights and time to evaluate our manuscript and for their constructive suggestions how to improve our work. Please find a detailed response below.

Reviewer #1 (Remarks to the Author):

This study from Erdmann-Pham and Batra describes a new computational method, HiDENSEC, for detection of rearrangements and copy number changes in patient tumor samples analyzed with Hi-C/Fix-C data. The study makes use of several novel computational strategies for identifying rearrangements and copy number changes in Hi-C data, and applied this tool to three patient samples with multiple sections to study the evolution of the karyotypes in these samples. On the whole, detection of rearrangements and copy number changes using Hi-C data has been demonstrated by several groups previously, and at least one study which the authors cite have performed such analysis on FFPE samples. Further, while it is interesting that they can find events disrupting NF1, CDKN2A, and PTEN, these are all pretty well known events in human cancers, and I think they are really underpowered to make meaningful conclusions regarding biological novelty from the 3 patient samples. The real technical novelty to me seems to be in their computational tool, for example in the estimation of tumor purity for Hi-C data. However, tools for estimating tumor purity have been established for WGS derived data, so the advancement introduced here is pretty modest. They also utilized several novel strategies to detect copy number changes and rearrangements. However, I find that they have not really extensively benchmarked their method against alternative approaches for detection rearrangements/CNVs, so the improvement in performance of their approach I don't think has been sufficiently demonstrated. While I find aspects of this study interesting, it seems somewhat preliminary in terms of novel biological insights or in the demonstration of improved computational ability for CNV/SV detection. I have divided my comments into major and minor comments below.

We thank the reviewer for their positive feedback about the novelty of analyzing complex mixed cancer samples. Responses to each of them are detailed in the following section, with explicit revisions of the manuscript referenced and often reproduced within the responses for clarity.

Major comments:

The authors need to improve their benchmarking against known gold standard and against other methods for detecting rearrangements and copy number changes. They compare with HiNT, but there are other methods for detecting rearrangements in Hi-C data (HiCtrans, EagleC, hic-breakfinder) that are untested. They also do not perform comparisons with other methods for detection of CNVs (though they cite several tools that are used to do so).

We have updated our methodological benchmarking to include comparisons with EagleC and hic_breakfinder (HiCtrans appears to suffer from deprecated dependencies, and so was omitted). We now include these results in the updated Figure 3:

This performance comparison reveals three directions in which HiDENSEC provides meaningful contribution alongside and beyond EagleC and hic_breakfinder:

1. While EagleC and hic_breakfinder's precision in calling rearrangement events rivals or slightly exceeds that of HiDENSEC for cases where the non-reference population is present at relatively large mixture proportions (50% and higher), this trend reverses for smaller sub-populations (30% and lower), where HiDENSEC matches or outperforms the two methods.
2. This emphasis on precision comes at a trade-off in recall: both EagleC and hic_breakfinder call events highly conservatively, resulting in 30% (for large mixture proportions) to 80% (for small mixture proportions) of true events being missed by these

methods under default significance thresholds. Relaxing significance thresholds for their methods relative to what their authors suggest resulted in little change to the number of called events. HiDENSEC, on the other hand, achieves consistently high recall in all situations, at the cost of calling two to three times as many events. Either behavior might be preferable depending on the application at hand.

3. The type of events HiDENSEC is designed to detect (large-scale genomic rearrangements associated with copy number variation) appear to be less reliably identified by EagleC and hic_breakfinder, suggesting that one of those methods in tandem with HiDENSEC may deliver complementary performance.

We have added elaborations along these lines in the section of the main manuscript Detecting reciprocal and copy-number altering translocations. We would also like to emphasize the capability of HiDENSEC to infer sub-population counts and their associated mixture proportions, which is absent from both EagleC and hic_breakfinder.

Copy number calls provided by HiDENSEC were assessed in comparison to both dSKY calls (see Figure 2a) and data from the UCSF500 gene panel (see Supplementary Figure 5), providing external validation in addition to the internal consistency that is confirmed in the *in silico* and *in vitro* mixture experiments (Figure 2b,d). These validation analyses were present in the original submission and we have further highlighted their appearance in the main manuscript. We use dSKY as our ground truth because by taking a haplotype-based approach, it is able to reliably characterize genome-wide as well as focal copy number changes present in a tumor genome across a range of tumor purities.

We share the reviewer's concern regarding discussion of, and benchmarking against, gold standard ground truth data genomics rearrangements, and discuss the subtleties we perceive to be inherent in such analysis in the following comment.

The methods for assigning "gold standard" SVs are pretty ad hoc. There are plenty of samples with high quality reference datasets in terms of whole genome sequencing, RNA-seq, or other methods they could use as an unbiased "gold standard."

External validation similar to our CNV benchmarking would indeed be desirable. However, there are reasons to question the accuracy of such standards as provided in the literature, as now discussed in the main text (page 7), which led us to manually curate lists of relevant events instead. More specifically, though existing ground truth labels appear to be of high precision, it is not clear that they are exhaustive enumerations of all large-scale rearrangements in the genome in question. For example, K562 (one of the most comprehensively and recently characterized cell lines for which data are available PMID: 30737237) exhibits the following two intensity patterns in its Hi-C matrix (figure layout as in Supplementary Figure 10):

Both of these cross-sub-regions are called by at least one method (HiDENSEC, EagleC, hic_breakfinder or HiNT), and each is strongly indicative of a true translocation (indeed, omission of these events leads to inconsistencies of the copy number profile). Yet neither of these two translocations are included in the published K562 “ground truth” referred to above. The probability of such intensity patterns occurring by chance is exceedingly small; coupled with the ability of Hi-C to directly probe physical proximity of two genetical locations casts doubt on how useful existing ground truth labels are for high-resolution method benchmarking. Therefore, we chose manual curation (which leads to a set of labels that is a superset of externally provided ones) to obtain ground truth for the HCC1187 cell line which we believe is highly representative of performance, instead of comparing on many benchmarks of uncertain completeness. Our benchmark set may be useful for other groups, and is provided in Supplementary Table 3.

We also considered using simulated Hi-C maps from pipelines such FreeHi-C (PMID: 31712779) and Sim3C (PMID: 29149264). However, matrices produced by Sim3C appear to generally lack complexities that are typical of real data, including intensity decays and both on- and off-diagonal confounding, as shown on two representative events in the following figure (the empirical, uncorrected copy number profile is displayed on the left, with insets of the two Hi-C submatrices on the right). This is now also noted in the main text.

The suitability of these simulated maps for benchmarking is thus unclear. FreeHi-C attempts to correct for these simplifications, but does not yet model structural variants of a given genome.

These observations have been added to the Detecting reciprocal and copy-number altering translocations section, and as Supplementary Figures 19 and 20.

Hi-C also has a known bias for detecting shorter rearrangements that are more readily detected by WGS, but there isn't a comparison of the ability of their method to

The primary goal of HiDENSEC is to identify large-scale genomic rearrangements spanning at more than 1 Mbp, which is a strength of Hi-C relative to WGS since it can measure contacts across extensive genomic stretches more explicitly (cf. our response to the comment immediately preceding this one). We agree that WGS is more conducive to characterizing variation below this scale, and have phrased this point more clearly in the Discussion section of the main manuscript.

They test the ability of the HiDENSEC method for recall, but not necessarily for false positives. The improvements seen in HiDENSEC (Fig. 3b) could be due to it being more "aggressive" in making calls. They should estimate the fraction of SV calls made by HiDENSEC that are "true" from gold standard datasets.

Figure 3a is not a conventional ROC plot, but differs from such in that its horizontal axis enumerates the top-ranked calls produced by each method (rather than the false positive rate). As such, both recall (on the vertical axis) and precision (as the fraction of step-increases up to a fixed number of calls) can be read from it. (We took the 'false positives' referred to in the reviewer's comment as querying precision rather than false positive rate, as the imbalance between events and non-events results in type 1 errors near zero for most reasonable methods.) The updated figure caption now emphasizes this important distinction.

Minor comments:

In the legend for Figure 1, they say that "FFPE samples may be microdissected." This is pretty vague. Do they perform microdissection? If so, say it and explain how. If not, this should be deleted (or moved to the discussion where speculative statements are more appropriate).

We thank the reviewer for this comment. We now have updated the information in the material and method section. This now reads: 'Tumor bearing areas for patient 1 and 2 were

microdissected from 10 μm thick unstained sections, using HE-stained sections as guidance. Samples for patient 3 were analyzed without microdissection.'

I think it is a bit of an overstatement to say that HiNT is the gold-standard for Hi-C analysis. Other newer methods, such as EagleC, have also been introduced that have very good performance.

The release of EagleC coincided with our analyses, and so was missed during the first round of benchmarking. Updated validation is now included in the revised manuscript (cf. our response to the first major comment above).

The authors state (line 124) that "A novel feature of our method is correcting the HiC signal for covariates including chromatin compartment and GC content, which allows accurate determination of copy number and tumor purity, and higher confidence detection of interchromosomal rearrangements." Correction for GC content has been described in numerous previous studies analyzing biases in Hi-C data (PMID: 22001755, PMID 23023982), so this isn't a particularly novel idea.

Thanks for making this point. We should have cited these papers for including GC content as a covariate. To our knowledge, however, ours is the first study to consider chromatin compartment in this manner, and we have clarified these points in the revised text.

The authors state (139) "The application of high-throughput chromatin conformation capture methods to FFPE samples (Fix-C) opens new possibilities for studying chromosome rearrangements in solid cancers." While I think that the Fix-C approach to studying FFPE samples is pretty cool, it isn't strictly necessary for studying solid tumors. Several previous studies have used solid tumor samples to generate Hi-C data previously (PMID 32841603, 28655341), so this is a bit of an overstatement.

Thanks for pointing out these two references, which we now cite. Both Harewood et al. (PMID: 28655341) and Johnstone et al. (PMID: 32841603) profiled genome topology in primary (i.e., fresh) colon tumor samples. The use of FFPE allows for retrospective studies of preserved samples, and also (by sectioning and microdissection) can begin to document spatial differences within a solid tumor. We have added these points to the discussion.

The authors state (line 225): "We therefore only used the presence/absence of these off-diagonal signals to detect large structural variants, and inferred absolute copy numbers and tumor purity based on the on-diagonal intensity analysis as described above (Supplementary Note 1)." My question related to this point is for balanced translocations. Can they detect absolute copy numbers under such circumstances that do not involve copy number changes? From the methods I get the impression that they don't do this (or at least from Supp. Fig. 4 believe it is not accurate), but to the best of my knowledge estimating absolute copy of balanced rearrangements is something that is not possible in WGS data, and this study would represent the first time I am aware of that such signals might be estimated.

The reviewer's understanding on the extent of HiDENSEC is correct: in its current formulation, HiDENSEC does not attempt to characterize absolute copy numbers of balanced translocations, but rather provide all copy number configurations that are consistent with the observed Hi-C signal (which in some cases may be unique). Narrowing these configurations to a more refined set may be possible through probing within-chromosome off-diagonal intensities more closely and/or analyzing sample frequency spectra as was performed on Sample 3. This is a direction for future work on HiDENSEC.

The Figure legend for Fig. 3a is a little confusing, particularly the meaning of the dashed versus solid lines. Which represents type 1 vs type 2 vs. all? Do they show both type 1 and type 2?

Solid lines measure recall with respect to all ground truth events, while dashed lines only consider the union of type 1 and type 2 events. This distinction is made more explicit in an updated figure caption.

They seem to have included a note to “[cite everyone]” in the introduction (line 91).

This citation is now populated with the appropriate references.

Reviewer #2 (Remarks to the Author):

This manuscript from Erdmann-Pham et al presents the development of HiDENSEC, a new tool that uses Hi-C data (Fix-C chromatin conformation capture protocol) to investigate genome evolution. Key strengths of HiDENSEC are an ability to extract useful information from samples that have been formalin fixed, paraffin embedded, identify chromosome rearrangements that are copy number altering and copy number neutral, maintain sensitivity even when cancer cell populations are diluted by normal cells, estimate tumor purity and absolute copy number, and determine allele frequencies. Other key strengths are an ability to map rearrangements at repetitive sequences like centromeres and telomeres. Benchmarking analyses indicate that HiDENSEC compares favorably to a similar method named HiNT. The authors use HiDENSEC to investigate chromosome rearrangement architectures across three melanomas and associated nevi. Interestingly, HiDENSEC powers the authors to identify telomere fusions in melanoma genomes, a difficult feat to accomplish using traditional short-read sequencing. Overall, the method development is clearly articulated and the experiments are rigorously performed. HiDENSEC represents an important new tool for researchers seeking to characterize cancer genomes across time in heterogeneous samples. There are no major surprises or insights gained into melanoma genome evolution described here, however this is a minimal limitation in an article focused on method development.

The authors compare HiDENSEC against HiNT, but do not extensively characterize HiDENSEC against more commonly used gold standards like paired-end sequencing approaches. A key potential advantage of short read sequencing is the ability to call base substitutions alongside chromosome rearrangements. I do not think further experimentation is necessary but it would be helpful to non-specialists to make room for these comparisons in the discussion section.

One minor note is that I spotted several typographical errors throughout the text. For example: 'Cite everyone' typo on line 91; Line 94 'decay' typographical error.

We thank the reviewer for positive feedback and helpful suggestions. We performed another round of checking for typographical errors, and added a paragraph to the Introduction section contrasting HiDENSEC with other Hi-C based large-scale rearrangement detection approaches.

Reviewer #3 (Remarks to the Author):

In this study, the authors present an approach that exploits Hi-C data generated from FFPE tumor samples to infer tumor purity, absolute copy number status, and complex rearrangements. The idea is overall interesting and well implemented with supporting results, although it feels as if several conclusions were rushed and substantial additional evidence should be shown to support these conclusions. Beyond my main comments listed below, the feeling of incompleteness is further conveyed by multiple imprecisions throughout the text, as if this was a draft and not yet a definitive version of the manuscript (see lacking or incorrect definitions, missing and misplaced references to figures, inconsistent annotations, etc.). I reported some of them at the end of this report, but I would strongly encourage the authors to carefully revise the text in its entirety.

We thank the reviewer for their helpful comments. Responses to each of them are detailed in the following section, with explicit revisions of the manuscript referenced and often reproduced within the responses for clarity. Moreover, we carefully revisited the text as a whole, corrected imprecisions, fixed incomplete statements and streamlined arguments.

Major comments:

1) The potential issue of tumor clonal heterogeneity is briefly touched upon by the authors but quickly dismissed saying that "it is typically sufficient to include one or sometimes two cancer genotypes along with the diploid reference genome" (from Supplementary Note 1). This is a bold statement and not necessarily supported by the literature where tumors exhibiting more than 2 clonal subpopulations have been reported in multiple contexts (see for example PMID:31209394, PMID:28445469, PMID:29656895 and many more).

Are there truly at most 2 clones in these samples or you cannot detect more than 1 or 2 given the resolution of the experiment? Indeed, the Hi-C experiments used in the manuscript seem to have an overall low number of reads: sample 2 I and II are both below 50 mln, which is very low considering these are admixes of normal and cancer cells, and only sample 3 I and II have more than 150 mln; coincidentally this is the sample where more than 1 clone is detected. The authors should introduce a control e.g. by performing whole genome sequencing of these samples and comparing the estimated clonality and/or by generating synthetic samples with more than 2 populations and testing the limit of detectability of their approach (e.g., combining 2,3,4,5,6 cell populations at different proportions). The latter should be doable considering the large amount of Hi-C datasets available for cancer cell lines.

Thanks for raising this point, which has both a biological and a statistical component.

From a biological perspective, it is certainly true that there can be more than one or two cancer genotypes in a tumor sample, and we did not intend to dismiss this possibility. We have added a note in the main text, and in Supplementary Note 1, to emphasize this point. We do document that our method can detect cell fractions of 10% or more but is less sensitive to rarer populations. Detectability of clonal populations will also depend on how different populations are from one another. We agree that deeper HiC sequencing could enable more sensitive detection

of additional populations, and have noted this in the Discussion and in the text that presents the HiDENSEC method.

The cited quotation from Supplementary Note 1 used “sufficient” in a technical statistical sense: we found that the copy number profiles of the specific samples analyzed in the main manuscript were statistically compatible with relatively few subpopulations. This may not be the case for other samples, which may require a model with more than 2 cancer subpopulations to achieve statistical sufficiency. HiDENSEC automatically assesses such statistical sufficiency, and returns the least number of subpopulations required for it. We thank the reviewer for raising this possible confusion, and have clarified this point in the revised main text with an additional paragraph; in the Discussion; and in Supplementary Note 1.

As outlined in sections 3 and 4 of Supplementary Note 1 (and the paragraph surrounding the quotation in question), we believe an approach based on statistical compatibility is necessary for two primary reasons:

1. From a theoretical perspective, the number of subpopulations is not “identifiable.” For example, the Hi-C map of a single subpopulation with many translocations is indistinguishable from a map that arises from the superposition of many subpopulations (one for each translocation), as long as each subpopulation’s mixture proportions are similar. Any method attempting to call subpopulations must necessarily resolve this ambiguity in some manner. HiDENSEC does so by appealing to parsimony, and seeks to characterize the smallest number of subpopulations that are required to produce the observed copy number profile.
2. Whether or not a given number of subpopulations is statistically compatible with an observed copy number profile will depend on the signal-to-noise ratio of the data. More specifically, the power to discern a subpopulation’s contribution to Hi-C intensities increases with the extent to which its mixture proportion differs from that of all other subpopulations (and from zero) relative to the stochastic fluctuations of Hi-C counts associated with the same effective ploidy. Thus, differences of mixture proportions that are of the same order as (or smaller than) the stochastic noise cannot be resolved by any Hi-C based method.

Accurately detecting distinct subpopulations is therefore a function of accurately detecting changes in effective copy numbers relative to noise, which in turn can be probed by varying the mixture proportion of a single subpopulation. This is the experiment we performed in Figure 2cd, and so similar resolution thresholds would be expected to apply to situations involving multiple subpopulations (that is, mixture proportion differences of ~10% in the samples analyzed in the main manuscript are reliably picked up by HiDENSEC).

The Online Methods and Results sections of the main manuscript have been updated to more transparently emphasize the points outlined above.

2) The authors have only compared their approach with HiNT, claiming this is the gold standard for these types of analysis from Hi-C data. While I can trust that, the truth is that for these analyses (purity, copy number status, and, especially, structural variants), the true gold standard

is still whole genome sequencing. A comparison of their results with those obtained by whole genome sequencing data is warranted. The authors may search and use available datasets where both information are accessible (these exist again for several cancer cell lines) and compare the copy number and structural variant calling ability.

We agree that the original formulation of our manuscript has been insufficiently clear about this concern. We have now added an updated benchmarking analysis including comparison with EagleC (PMID: 35704579) and hic_breakfinder (PMID: 30202056). We have also added a detailed discussion of the subtleties of comparing against existing “gold standard” protocols; in particular we note that there are clear rearrangements found in benchmark cancer cell types that are not included in the “true” feature set. Please consult our responses to the major comments 1 and 2 of Reviewer 1 for a summary of these updates.

3) Another undiscussed issue concerns compartment differences among different cell lines. First of all, although never explicitly mentioned, based on supplementary information I'm guessing that the authors are using the compartment calls generated in (Rao et al. 2014 - PMID:25497547) for the GM12878 cell line. Is this correct?

(I assumed so because in the supplementary note the authors mentioned the use of 6 sub-compartments, which is the number proposed in that study, although the labeling is inconsistent with that proposed in that study, it should be A1, A2, B1, B2, B3, and B4. Unless of course this classification comes from somewhere else.)

The problem is that different cell types can have very different compartments and while on a global scale the correlation will still be high, I wonder whether local differences at specific loci may affect copy number calls at that specific loci. This should be tested.

This is of course a complex issue because in the admix of multiple cell types (e.g., normal and cancer)

you will have different compartments being admixed from highly heterogeneous immune, stromal, and cancer cells.

A possible solution could be regressing only with respect to loci that have been shown have highly consistent compartment assignments across multiple cell types (e.g., as shown in PMID:33972523 - data should also be available). The authors should test this possibility, even if in their samples it won't change the conclusions, ignoring compartment differences is a risk.

We obtained subcompartment information for GM12878 from the SNIPER manuscript (PMID: 31699985), which is based on data from Rao et al. 2014 - PMID:25497547. We introduced a dummy subcompartment denoted by A0 corresponding to all the loci in the genome where subcompartment information was not available. Additionally, since the authors of SNIPER note that loci corresponding to B4 subcompartments in GM12878 are only present in chromosome 19 spanning less than 0.4% of the genome, the B4 subcompartment is missing from the analysis.

We agree that the question of robustness against compartment misspecification is an interesting and important one. After exploring various directions to possibly resolve this concern, we have

come to the opinion that the currently implemented covariate correction (based on compartment calls of SNIPER on GM12878) strikes a favorable balance between various trade-offs:

1. The compartment calls produced in SNIPER appear to generally separate Hi-C intensities more robustly than alternatives. E.g., the following figure compares ploidy-corrected compartment specific Hi-C counts across all the samples (each subplot representing a sample) that were investigated throughout our analysis (including cell lines) for compartments produced by either SNIPER (top panels) and PMID:33972523 (the reference mentioned in the reviewer's comment, which implements a method called Calder). While neither of the samples exhibit as clear separability as GM12878 on the conventional compartment calls, the amount of separability still exceeds that of Calder-compartments substantially, suggesting generally greater efficiency when correcting covariates.

2. This increased separability remains true in particular when comparing Calder compartments of high inter-cell-line consistency (namely, B22 and A11) against the equivalent compartments produced by SNIPER (B3 and A1), as depicted by the following figure.

3. Conditioning only on the highly conserved A11 and B22 compartments subsamples the usable data to about 28% of its original size. Apart from the associated increase in uncertainty around continuous quantities like inferred copy number profiles and mixture

proportions, such subsampling is prone to removing precise breakpoint locations which are crucial for accurately proposing and characterizing off-diagonal events. The following figure showcases how this would play out in HCC1187, with filled areas indicating conserved compartment structure:

4. The above three conclusions were obtained from samples analyzed in this study, which naturally raises the concern to what extent they remain applicable to Hi-C maps not investigated here. Because of the subtleties involved in curating ground truth data (as elaborated on in our response to comment 2 of reviewer 1), comparing compartment suitability on a large set of data resources is difficult to perform in an automated fashion, which motivated a search for alternative ways in which HiDENSEC might guard against compartment misspecification:
 - a. Though SNIPER compartment designations are employed by default, the user now has the option to provide compartment calls of their own if they believe those to be more accurately representative of the sample in question. Moreover, Hi-C on-diagonal profiles can be subsampled to include only counts associated with certain compartment structures (e.g., those shown to be more conserved).
 - b. HiDENSEC now offers to perform within-sample covariate correction, in that a corrector function g may be trained on the sample in question (rather than needing to obtain it from a separate diploid Hi-C map acting as reference). To do so, it proceeds by isolating on-diagonal counts likely associated with the same ploidy, and uses only these isolated counts to regress out covariate effects. Although the so-performed regression will generally draw from smaller sample sizes and compartment mislabelling at training (both of which will impact uncertainty quantification), it protects against compartment misspecification through implicitly reducing bias.

Supplementary Note 1 now contains an extended discussion detailing these considerations.

4) This might be a minor point but I wonder why the authors do not try to infer the "purity" of translocations/re-arrangements. They say that doing this at off-diagonal entries is difficult and subject to biases. But bins around the breakpoint (or even where the breakpoint is) are only seemingly 'off-diagonal' but they are actually on the diagonal of the chimeric chromosome, so it should be possible to apply the same approach. This could be interesting if it allows to estimate the fraction of cancer cells exhibiting a given variant.

Even bins immediately around the breakpoint (corresponding to a window size of 1 in Supplemental Figure 4) appear to exhibit behavior that is distinct from proper on-diagonal elements (as is illustrated by the same Supplemental Figure), and so render ploidy inference difficult. As alluded to in the main manuscript, we conjecture that these differences in behavior might be a result of:

1. Breakpoint occurring within bins (associated with the discretization of Hi-C matrices at 50kb resolution) rather than at their boundaries: such occurrences will artificially reduce observed contact intensities (compared to proper on-diagonal bins) between the bin containing the breakpoint and its neighbors. The extent of this reduction depends on the precise location of the breakpoint within the bin.
2. Chimeric chromosomes exhibiting distinct chromatin folding compared to unaltered chromosomes: fusing components of two distinct chromosomes appears to result in less tightly packed chromatin structure than what would be expected when these components belong to the same chromosome. The extent to which chromatin folding is impacted appears to depend on the donor chromosomes, and thus may be difficult to account for quantitatively.

Given that the effective ploidy of translocations and rearrangements is often determined entirely by mixture proportion and marginal copy number context (that is, the marginal copy numbers of the involved chromosome segments and their fusion partners), HiDENSEC does not currently employ a more detailed off-diagonal analysis. However, exploring these signals further to disentangle ambiguity where it arises or to further understand folding structure of chimeric chromosomes is an exciting prospect, and we thank the reviewer for bringing it to our attention.

5) The authors introduced multiple tumor phylogenies and genotype reconstructions based on what is stated to be a maximum parsimony approach, but it looks like this was all done by manual curation. Is that so? The authors should elaborate more on the procedure that was followed and if these results come from their intuition and common sense, it should be said explicitly (and possible alternative approaches should be tested).

Every configuration of copy number profile and translocation/rearrangement events inferred by HiDENSEC is compatible with a certain set of genotypes. For samples 1 and 2, these sets of compatible genotypes are singletons (subject to the parsimony assumptions which are mentioned in the comment at hand, and which are related to our discussion of the reviewer's first comment on multiple subpopulations), and so required no further analysis. Sample 3, on the other hand, admits multiple genotypes consistent with the observed data, and thus necessitates follow-up inspection to disambiguate the role of each genotype candidate. This was achieved using data from UCSF500 gene panel sequencing (Figure 6e) in combination with FISH analysis (Figure 7e). Similarly, sample phylogenies were reconstructed using information from the same UCSF500 data. For samples 1 and 2, phylogenies are uniquely identified by this information, while for sample 3 all but three phylogeny candidates are eliminated. The phylogeny featuring the least number of independent duplicate events was selected for presentation in the main manuscript, with the remaining two compatible phylogenies listed in

Supplemental Figure 8. The corresponding figure captions and associated main text pointers have been expanded to elucidate this process more clearly.

6) Lastly, the discussion currently reads more as a summary of the results rather than a true discussion. Can the authors elaborate more on the current limitations, possible improvements, and broader applicability (e.g. compared to WGS) of their approach? I would further discussed for example the limited resolution of the Hi-C datasets, is this intrinsic of working with FFPE samples or could it be improved with further sequencing?

We thank the reviewer for this comment. We have updated the discussion to better outline the strength and limitations of our method.

Minor imprecision in the text (here I indicated a few but there are likely more)

- line89: Hi-C actually stands for 'high-throughput chromosome conformation capture'

This has been corrected.

- line91: "[cite everyone]" I believe there is a missing citation here?

The proper references have been added to the citation.

- line94: "decay slowly" - I would not define an exponential decay as 'slowly', what would be fast then?

This has been updated.

- line107: "the current gold-standard for analysis of Hi-C in data" what does that mean? I believe something is missing in this sentence

This has been corrected in the revised paragraph.

- Figure 1 is not referenced in the text

It is now cited.

- line 248: from the description there are 8 samples not 9

This has been corrected.

- suppl. fig 7 only shows schematics of the translocations for patient 2, and it points to suppl. fig 9, but here the legend refers to patient 3 - which is it?

This has now been corrected.

- line 319 and paragraph below: figure references to Fig 6 and Fig 7 seem misplaced and/or missing

This has now been corrected by adding more references to these figures.

- in the online methods, covariate correction equation has black boxes

We thank the reviewer for bringing this to our attention. We will work with the journal upon acceptance to make sure that the equations render correctly. For now, both those black boxes correspond to empty strings and hence we don't lose any information due to them.

REVIEWERS' COMMENTS

Reviewer #1 (Remarks to the Author):

Erdmann-Phamm, Batra et al. present a revised version of their manuscript describing the development of HiDENSEC, a method they have developed for analyzing rearrangements and copy number changes in heterogeneous Hi-C tumor samples. Compared with the original submission, I find the revision to be much improved. I have to admit, I was skeptical of the utility of the HiDENSEC tool from the original submission, but the comparisons the authors present against EagleC and Hi-C break finder have gone a long way of convincing me of its utility. It clearly has value and I suspect will be appreciated by the community. With this in mind, I find the manuscript essentially suitable for publication. There are still a few very minor points that I have listed below that I hope the authors can address. These are things that I think of almost as “take it or leave it,” as in if the authors feel strongly against addressing them I don’t think it would preclude publication. I list them here as I genuinely believe they would improve the manuscript (and they are quite minor). I appreciate the work the authors have put in and congratulate them on an interesting paper.

Minor comments:

They use the phrase “non-reference populations” for the mixture analysis, but to me this is unnecessarily confusing. I think they should just say “tumor”.

They mention that some of the events in K562 are not called in prior ground truth data sets (lines 284-292). One possibility the authors don’t mention would be an “indirect join” (I’m not sure if that is a real term). The idea would be you would have a complex event, with three loci that would be discontinuous in the reference that are joined together (A-B-C). It seems to me that the Hi-C data might also identify signal between A and C but that this could be missed by methods that directly test for the junction (WGS). On some level this is a true event missed by tools that may not detect complex events, but it is also sort of a false positive as it doesn’t represent a direct genetic junction.

The authors write: “HiCTrans was excluded due to difficulties with deprecated dependencies” They can chose to ignore this comment but it seems to me to be

unnecessary to include this. It seems like it would be too much to ask of people to list not only what they do compare and but also things they didn't do as well.

Reviewer #2 (Remarks to the Author):

The authors have satisfactorily addressed my previous concerns and I am in favor of publication of this revised manuscript.

Reviewer #3 (Remarks to the Author):

The authors have responded to all my concerns, although in some instances by simply amending the text rather than introducing new analyses. This is particularly true for my first and second comment.

In particular, it seems that all 3 reviewers pointed out the need to compare their calls for structural variants with the gold standards based on whole genome sequencing (Reviewer 1 and myself requesting proper comparisons). I appreciate the argument made by the authors that WGS is prone to error as well, but so is a manual curation based on visual examination of Hi-C maps (which is also subjective). Given the issue was brought up by multiple reviewers I would have expected a more thorough investigation, including proper comparisons using multiple cell lines, quantification of the differences between Hi-C based and WGS based approaches, and then highlighting in specific instances when such differences reflect a better accuracy of Hi-C or WGS approaches. I think these arguments are required if want to convince the community to use Hi-C rather WGS to call structural variants.

However, if the other reviewers are satisfied with the response provided, I will be as well.

REVIEWER COMMENTS

We would like to thank the reviewers once more for their valuable suggestions and comments to improve our work. Any follow-up reviews are responded to below.

Reviewer #1 (Remarks to the Author):

Erdmann-Phamm, Batra et al. present a revised version of their manuscript describing the development of HiDENSEC, a method they have developed for analyzing rearrangements and copy number changes in heterogeneous Hi-C tumor samples. Compared with the original submission, I find the revision to be much improved. I have to admit, I was skeptical of the utility of the HiDENSEC tool from the original submission, but the comparisons the authors present against EagleC and Hi-C break finder have gone a long way of convincing me of its utility. It clearly has value and I suspect will be appreciated by the community. With this in mind, I find the manuscript essentially suitable for publication. There are still a few very minor points that I have listed below that I hope the authors can address. These are things that I think of almost as “take it or leave it,” as in if the authors feel strongly against addressing them I don’t think it would preclude publication. I list them here as I genuinely believe they would improve the manuscript (and they are quite minor). I appreciate the work the authors have put in and congratulate them on an interesting paper.

We thank the reviewer for their constructive suggestions that helped improve the paper to its current form.

Minor comments:

They use the phrase “non-reference populations” for the mixture analysis, but to me this is unnecessarily confusing. I think they should just say “tumor”.

We agree with this comment and have made this correction.

They mention that some of the events in K562 are not called in prior ground truth data sets (lines 284-292). One possibility the authors don’t mention would be an “indirect join” (I’m not sure if that is a real term). The idea would be you would have a complex event, with three loci that would be discontinuous in the reference that are joined together (A-B-C). It seems to me that the Hi-C data might also identify signal between A and C but that this could be missed by methods that directly test for the junction (WGS). On some level this is a true event missed by tools that may not detect complex events, but it is also sort of a false positive as it doesn’t represent a direct genetic junction.

We thank the reviewer for pointing out this possibility, which indeed raises the question of what type of events should be treated as true positives: on the one hand, detecting indirect joins might be desirable as it aids in the detection and reconstruction of such complex

rearrangements, while on the other hand it ought not be mistaken for a direct join. We now mention this scenario in the 'Detecting reciprocal and copy-number-altering translocations' section of the paper, where we also point out that the two events detected by Hi-C based methods but absent from WGS-references are not of this complex type (which can be checked as any 'linking locus B' should leave a Hi-C signal in the event-associated rows of the intensity matrix). That is, though some events recognized by Hi-C might be attributable to these types of complex joins, at least a subset of them appears to correspond to direct joins that remain undetected by WGS-based procedures.

The authors write: "HiCTrans was excluded due to difficulties with deprecated dependencies" They can chose to ignore this comment but it seems to me to be unnecessary to include this. It seems like it would be too much to ask of people to list not only what they do compare and but also things they didn't do as well.

We agree with this comment and have removed the phrase in question.

Reviewer #2 (Remarks to the Author):

The authors have satisfactorily addressed my previous concerns and I am in favor of publication of this revised manuscript.

We thank the reviewer for their constructive suggestions that helped lead to the current manuscript.

Reviewer #3 (Remarks to the Author):

The authors have responded to all my concerns, although in some instances by simply amending the text rather than introducing new analyses. This is particularly true for my first and second comment.

In particular, it seems that all 3 reviewers pointed out the need to compare their calls for structural variants with the gold standards based on whole genome sequencing (Reviewer 1 and myself requesting proper comparisons). I appreciate the argument made by the authors that WGS is prone to error as well, but so is a manual curation based on visual examination of Hi-C maps (which is also subjective). Given the issue was brought up by multiple reviewers I would have expected a more thorough investigation, including proper comparisons using multiple cell lines, quantification of the differences between Hi-C based and WGS based approaches, and then highlighting in specific instances when such differences reflect a better accuracy of Hi-C or WGS approaches. I think these arguments are required if want to convince the community to use Hi-C rather WGS to call structural variants.

However, if the other reviewers are satisfied with the response provided, I will be as well.

We appreciate, and to a large extent share, the reviewer's concern about adequately comparing methods based on Hi-C and WGS, respectively. We agree that a more thorough understanding of the differences between these two approaches is desirable, though believe that such an analysis would quickly escape the scope of the current paper. HiDENSEC's primary utility lies in providing an integrated tool for inferring subpopulations, mixture proportions and large-scale rearrangements, whose individual components compare favorably with existing tools by our assessment (which is a sentiment that seems to be shared by the other two reviewers). Characterizing differences in Hi-C and WGS at large would likely require exploiting multiple such tools (given, e.g., the complementarity of HiDENSEC with EagleC/HiC-breakfinder that we have observed) and resolving several subtleties (like the precise interpretation of differences in detected events: e.g., determining whether an intensity pattern appearing in Hi-C but not recognized by WGS represents a true or false positive appears to again require some subjectivity) that do not directly pertain to method development, and thus has not been pursued in this work. We are looking forward to future investigations tackling this task, and hope that HiDENSEC might be of use in these endeavors.